



# Aircraft Engine Dust Ingestion at Global Airports

Claire L. Ryder[1], Clèment Bézier[1,2], Helen F. Dacre[1], Rory Clarkson[3], Vassilis Amiridis[4], Eleni Marinou[4], Emmanouil Proestakis[4], Zak Kipling[5], Angela Benedetti[5], Mark Parrington[6], Samuel Rémy[7], Mark Vaughan[8]

[1]Department of Meteorology, University of Reading, Earley Gate, PO Box 243, Reading, RG6 6BB, UK
[2]Service des Avions Français instrumentés pour la Recherche en Environnement (SAFIRE), Météo-France, CNRS, CNES, Toulouse, France
3 Rolls-Royce plc, Derby, UK
4 National Observatory of Athens, IAASARS, Athens, 15236, Greece
5 ECMWF, Shinfield Park, Reading, RG2 9AX, UK
6 ECMWF, Robert-Schuman-Platz 3, 53175 Bonn, Germany
7 Hygeos, Lille, France
8 NASA Langley Research Center, Hampton, VA, USA

*Correspondence to*: Claire Ryder (c.l.ryder@reading.ac.uk)

**Abstract.** Atmospheric mineral dust aerosol constitutes a threat to aircraft engines from deterioration of internal components. Here we fulfil an outstanding need to quantify engine dust ingestion at worldwide airports. The vertical distribution of dust is of key importance since ascent/descent rates and engine power both vary with altitude and affect dust ingestion. We use representative jet engine power profile information combined with vertically and seasonally varying dust concentrations to calculate the 'dust dose' ingested by an engine over a single ascent or descent. Using the Copernicus Atmosphere Monitoring Service (CAMS) model reanalysis, we calculate climatological and seasonal dust dose at 10 airports for 2003-2019. Dust doses are mostly largest in summer for descent, with the largest at Delhi (6.6 g). Beijing's largest dose occurs in spring (2.9 g). Holding patterns at altitudes coincident with peak dust concentrations can lead to substantial quantities of dust ingestion, resulting in a larger dose than the take-off, climb and taxi phases. We compare dust dose calculated from CAMS to spaceborne lidar observations from two dust datasets derived from the Cloud–Aerosol Lidar with Orthogonal Polarization (CALIOP). In general, seasonal and spatial patterns are similar between CAMS and CALIOP though large variations in dose magnitude are found, with CAMS producing lower doses by a mean factor of 2.4±0.5, particularly when peak dust concentration is very close to the surface. We show that mitigating action to reduce engine dust damage could be achieved, firstly by moving arrivals and departures to after sunset and secondly by altering the altitude of the holding pattern away from that of the local dust peak altitude, reducing dust dose by up to 44% or 41% respectively.



# 1    1 Introduction

Aircraft gas turbine engines ingest ambient atmospheric gases and aerosols in addition to pure air. Many atmospheric components cause damage to internal components of aircraft engines, through erosion, corrosion or deposition (Clarkson, 2019). Since the volcanic eruption of Eyjafjallajökull in 2010, which closed down most European airspace for over 5 days and caused \$5 billion in economic losses (Prata and Rose, 2015; Prata et al., 2018), renewed focus has been given to the impact of volcanic ash on aircraft engines.  High concentrations of volcanic ash can compromise safety and thus the highest ash concentrations must be avoided (Clarkson et al., 2016). Mineral dust, originating from arid regions and uplifted by strong surface winds (Knippertz and Stuut, 2014), also causes damage to aircraft engines (Clarkson, 2019). It is generally unlikely to be a safety issue in itself, instead causing engine components to degrade more rapidly, impacting efficiency and maintenance costs. Indirectly, icing caused by mineral dust particles acting as ice nuclei can be a serious threat to aviation (Nickovic et al., 2021). Despite this, under extremely dusty conditions, operations are more likely to be cancelled due to visibility reductions rather than risk to aircraft engines (Middleton, 2017).

In recent years the potential for dust-related engine damage has increased as a result of increased flight operations in dusty regions and greater susceptibility of engine parts to dust damage. This is firstly, due to a move towards more fuel-efficient engines which employ hotter, higher pressure cores with tighter clearances, more complex cooling systems and more sophisticated protective coatings, resulting in engines which are less tolerant to dust as well as other atmospheric contaminants (Clarkson, 2019). Secondly, air traffic activity has significantly increased in arid, dusty areas (particularly the Middle East), increasing aircraft exposure to mineral dust (O'connell and Bueno, 2018). Increased air traffic worldwide means that infrequent but large dust events are also more likely to impact air traffic. Thirdly, there has been a shift away from airlines directly funding engine repair and overhaul – under time and material arrangements with overhaul bases – to service contracts where they pay an overhaul provider (usually the engine manufacturer) a fee per flight hour, transferring the repair and maintenance liabilities to that overhaul provider. Combined with increased economic pressure on airlines to maximise operations, this has resulted in increased likelihood for aircraft to operate in dusty conditions, especially as dust does generally not represent an aviation safety threat.

The amount of dust ingested by aircraft engines depends on both the total dust concentration, its vertical distribution, and spatial and temporal variations as well as the engine power which varies with time and altitude. There is an outstanding need to quantify aircraft dust dose (the total dust mass ingested over a given amount of time), particularly due to the vertical distribution of dust, since this varies spatially, seasonally, and diurnally depending on location and dominant meteorology in uplifting and transporting dust plumes. Since different aircraft will ingest differing amounts of air (and therefore dust) during different phases of aircraft ascent and descent as a result of varying engine power and duration (Clarkson, 2020), the vertical distribution of dust will have a key influence on engine dust ingestion.



Model reanalyses of atmospheric composition, which include mineral dust aerosol (e.g., Inness et al. (2019)), are a powerful
tool in quantifying dust concentrations and their variability in space and time. Bojdo et al. (2020) used ECMWF hindcasts
from the Copernicus Atmosphere Monitoring Service (CAMS) to calculate engine dust dose at Doha airport for an Airbus
A380-841 with Rolls-Royce Trent 900 engines. They found that the average dust dose ingested into the engine core per
flight was 8.5 g, with peak dust ingestion occurring just after take-off and during aircraft transition into climb phase at
around 1 km. Additionally, they found that a twenty minute hold phase over the dusty Persian Gulf at around 3 km
accumulated a dose of 8 g. They focused their study on three dusty months at Doha at 6 h temporal resolution. However, an
assessment of engine dust dose at a wider range of airports, over a larger temporal range, and a higher sampling rate of
diurnal variability has not yet been undertaken.

Accounting for the vertical distribution of dust is vital in calculating dust dose and for models of particle damage to engine
components (Ellis et al., 2021; Bojdo et al., 2020). Spatial distributions and vertically resolved dust concentrations are
available from atmospheric models incorporating mineral dust schemes, such as the CAMS forecasts and reanalysis.
Although CAMS has the advantage of assimilating total aerosol optical depth retrieved from satellite observations, there are
no constraints on how this is proportioned across different aerosol types or in the vertical dimension. Many satellite sensors
measure aerosol optical depth, but there are challenges in proportioning this into dust-specific components and little-to-no
information on the vertical distribution. One exception to this is spaceborne and ground-based lidars, which provide
vertically resolved profiles of dust properties. The importance of height-resolved dust information from ground-based lidars
has been highlighted for near real-time aviation warnings (Papagiannopoulos et al., 2020). Lidar observations from the
spaceborne Cloud–Aerosol Lidar with Orthogonal Polarization (CALIOP) have proved invaluable in characterizing the
vertical distribution of dust and its regional variations (e.g. Liu et al. (2008a); Yang et al. (2013); Song et al. (2021)).

Here we use time-varying three-dimensional dust concentration from the CAMS reanalysis over a 12 year period to
determine climatological engine dust dose for a range of worldwide airports, for a single ascent or descent manoeuvre (i.e.,
departure or arrival). We investigate the dust vertical profiles in different regions which occur due to varying regional
meteorology and how these affect dust dose, as well as the contribution from seasonal and diurnal variability in vertical
profiles. Finally, we compare the reanalysis dust profiles and associated dust dose to observational spaceborne lidar
retrievals of vertically resolved dust concentration from two CALIOP dust datasets. Section 2 describes the method, datasets
and dose calculations, Section 3 presents results of vertical profile dust climatologies and dose climatologies, Section 4
provides potential dose mitigation methods, Section 5 concludes.



## 2 Methods

### 2.1 Selection of airports

This work focuses on a selection of ten airports, selected based on air traffic activity levels, proximity to dusty regions and anecdotal reports of dust ingestion. The airports selected are Phoenix, the Canary Islands, Marrakesh, Niamey, Dubai, Delhi, Bangkok, Hong Kong, Beijing and Sydney, as shown in Figure 1. Broadly, The African and Asian airports are located in or around the edges of the well-known 'dust belt,' stretching from western Africa through to Northern China (Prospero et al., 2002). Singapore, Hong Kong and Bangkok are located south of major dust transport pathways but serve as major airports in the region and aircraft have reported occasional dust damage here. Sydney and Phoenix can be affected by large but infrequent dust storms originating from arid areas in eastern Australia and the western US (Ginoux et al., 2012). Due to the model resolution, the Canary Islands covers all airports on the islands.

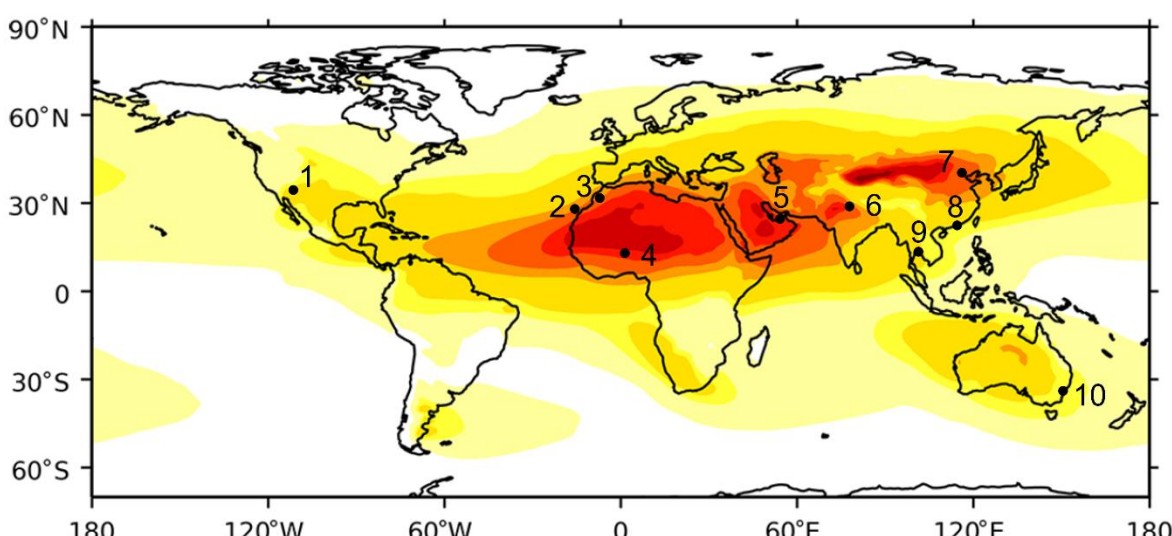

**Figure 1: Airport locations used in this study. 1= Phoenix, 2=Canary Islands, 3=Marrakesh, 4=Niamey, 5=Dubai, 6=Delhi, 7=Beijing, 8=Hong Kong, 9=Bangkok, 10=Sydney. Underlying contours show distribution of annual average column dust loading from CAMS from 2005-2014, adapted from Zhao et al. (2022).**

### 2.2 CAMS Reanalysis Dust Concentrations

We utilize the global Copernicus Atmosphere Monitoring Service (CAMS) reanalysis dataset (Inness et al., 2019) to estimate climatological, vertically resolved dust mass concentrations at each airport. Aerosol and meteorological modelling and data assimilation are carried out by the ECMWF Integrated Forecast System within the CAMS framework. Total aerosol optical depth (AOD, covering all aerosol species) from satellite retrievals are assimilated into CAMS, and concentrations of individual aerosol species are adjusted according to the modelled proportions and vertical distributions of each aerosol type, of which mineral dust constitutes one of five (Morcrette et al., 2009). CALIOP data are not included in the assimilation.



Thus, the CAMS dataset benefits from an observational constraint of AOD, affecting total aerosol column load, though how this is proportioned amongst different aerosol types and their vertical distribution is dependent on the model-simulated values. Dust emission is driven by modelled surface wind speed, soil moisture, surface albedo and bare soil fraction

conditions (Bozzo et al., 2020; Morcrette et al., 2008; Remy et al., 2019). Dust is represented by three size bins with upper/lower diameters of 0.06, 1.1, 1.8 and 40 µm, and transport following emission is controlled by model winds and associated meteorology.

We analyse data from January 2003 to December 2019, though in comparisons to spaceborne lidar we restrict this to 2007-

2019 to be consistent with that data. The nearest CAMS grid box (size of 80 km) to the location of each airport is selected. Dust mass mixing ratios are available at 3 hourly time intervals with 60 vertical levels of which we use 23, corresponding to the surface to approximately 6 km altitude. Total dust concentrations are calculated as the sum of dust over the 3 size bins, and by converting mass mixing ratios (kg/kg) to dust mass concentration (kg/m$^3$) as a function of altitude using standard meteorological conversions. Climatologies are calculated as monthly mean profiles for each airport using 3 hourly output

data from CAMS.

## 2.3  CALIOP Spaceborne Lidar

As an observational comparison to the CAMS dust profiles, we use spaceborne lidar data from the Cloud–Aerosol Lidar with Orthogonal Polarization (CALIOP) instrument aboard the Cloud–Aerosol Lidar and Infrared Pathfinder Satellite Observations (CALIPSO) satellite (Winker et al., 2010). CALIOP is an elastic backscatter lidar, providing information on

the spatial and vertical distribution of aerosols at wavelengths of 532 nm and 1064 nm. Attenuated backscatter profiles are measured, from which aerosol extinction profiles are derived, where extinction is the absorption and scattering coefficient at each height, broadly indicating the amount of aerosol in the atmosphere. Additionally, CALIPSO measures the linear depolarization ratio at 532 nm, enabling identification of dust aerosols, which are strongly depolarizing due to their non-spherical shape, in contrast to other aerosol types (Kim et al., 2018). CALIOP therefore provides an observationally driven

estimate of the vertical distribution of dust, although limitations exist and certain assumptions are required, as described below. The CALIPSO orbit track repeats every 16 days, (until 2018 when the orbit was adjusted), though global coverage is never achieved due to the small footprint size. Here we analyze two different 532 nm CALIOP datasets: the standard Level 3 dataset, produced and distributed by NASA's CALIPSO project, (henceforth referred to as 'CALIOP L3') and the LIdar climatology of Vertical Aerosol Structure (LIVAS) dataset, created from the CALIPSO measurements (Amiridis et al., 2015;

Amiridis et al., 2013). For purposes of comparison to CAMS, the CALIOP datasets are both reduced to the same vertical resolution of CAMS. For both datasets we use night time data only in order to retrieve maximum lidar signal-to-noise ratios in the absence of solar radiation. For comparisons between the lidar datasets and CAMS, CAMS data is also restricted to night hours and dates of the coincident overpass for consistency. We now describe the different processing between the CALIOP L3 and LIVAS datasets.



### 2.3.1 CALIOP L3

We use the Level 3 (L3) version 4.20 monthly mean 'Tropospheric Aerosol Cloud Free' product provided by NASA (Tackett et al., 2018), which currently provides the longest observational record of the vertical distribution of speciated aerosol. L3 data are derived from the Level 2 (L2) V4.20 data product where 532 nm integrated attenuated backscatter, estimated particulate depolarization, altitude, location, and surface type are used to identify the dominant aerosol type in a layer. In particular, if the layer-integrated depolarization ratio exceeds 0.20 (indicating non-spherical particles), then the layer is identified as 'dust.' A fixed lidar ratio of 44 sr is used to convert the measured dust attenuated backscatter coefficient profiles to extinction coefficient profiles (Kim et al., 2018, Young et al., 2018). This lidar ratio value is in agreement with the averaged ground-based lidar ratio measurements for desert dust, recently reported by (Floutsi et al., 2022), despite the regional dependence. L2 data are averaged and quality controlled to produce the L3 product, which provides monthly mean extinction profiles as a function of aerosol type (Tackett et al., 2018). CALIOP L3 data are produced on a 2°× 5° grid with a vertical resolution of 60m, up to 12 km altitude. We use the extinction coefficient at 532 nm profiles of atmospheric features classified as 'dust' aerosol. We use cloud-free conditions for night time CALIPSO overpasses only, in order to minimize uncertainties from cases contaminated by clouds overlying dust layers and by high sunlight illumination conditions (low signal to noise ratio). We exclude the 'polluted dust' and 'dusty marine' aerosol types. In very dusty cases, in-layer attenuation may mean that the CALIPSO feature detection algorithm does not detect the entire vertical extent of a dust layer, and the layer base will be assigned an altitude 90 m above Earth's surface. Data beneath this altitude (or layer) do not contribute to the summed extinction coefficient profile or sample count of atmospheric layers classified as 'dust' in terms of aerosol-subtype (Tackett et al., 2018). Each CALIOP L3 airport gridbox is selected to encompass the equivalent CAMS gridbox. Although the CALIOP gridbox is larger than that from CAMS, the results showed negligible sensitivity to including multiple CAMS gridboxes for the comparison. We use data covering 2007 to 2019.

### 2.3.2 LIVAS

We use the 'pure dust product' from the LIdar climatology of Vertical Aerosol Structure (LIVAS, Amiridis et al. (2015)). The LIVAS pure-dust dataset (Amiridis et al., 2013) is established based on CALIOP L2 V4.2 aerosol and cloud profiles and findings from the European Aerosol Research Lidar Network (EARLINET, Pappalardo et al. (2014); last access: 23/08/2022). LIVAS was developed to provide pure-dust and non-dust components of atmospheric scenes under the assumption of external aerosol mixtures (Tesche et al., 2009). The LIVAS 'pure-dust' product applies CALIOP L3 quality assurance parameters to height-resolved particulate depolarization ratio and backscatter coefficient profiles at 532 nm (Konsta et al., 2018) to decouple the contributions from non-spherical (dust) aerosol and other (non-dust) spherical aerosol components of the backscatter profile (Amiridis et al., 2013; Kim et al., 2018). LIVAS analyses are limited to those CALIOP L2 profiles already classified by CALIOP as either 'pure dust,' 'dusty marine,' or 'polluted dust' (Kim et al., 2018). If the particulate linear depolarization retrieved by CALIOP in any vertical bin is greater than 0.31 for pure dust (Floutsi et al.,





2022), the whole volume in that vertical bin is considered dusty; if it lies between 0.05 to 0.31 then the proportion of backscatter derived from dust aerosol is calculated. Through the sum of these signals, the profile of 'pure-dust' backscatter coefficient at 532 nm. A region-specific lidar ratio (see supplement for values) is then used to calculate 'pure-dust'
extinction coefficient profiles from the backscatter coefficient profiles (Amiridis et al., 2015; Marinou et al., 2017; Proestakis et al., 2018; Floutsi et al., 2022). Here we use the LIVAS 'pure dust extinction coefficient product' for cloud-free, night-time cases, on an identical 2°×5° grid to match the CALIOP L3 data, for the years 2007 to 2019. In cases where data is affected by a strong return signal from the ground in the lowest two altitude bins, a value from the first altitude above 100m is replicated downwards.

### 2.3.3 Differences between CALIOP L3 and LIVAS

In summary, both CALIOP L3 and LIVAS use CALIOP L2 V4.2 profiles. Both approaches use a particulate depolarization threshold to identify dust profiles, though in each dataset this is applied in very different ways. The CALIOP L2 analyses retrieve aggregate extinction coefficients from detected aerosol layers based on a single aerosol subtype classification (one of which is 'dust') applied to the entire layer. For heterogeneous layer types (e.g., dusty marine and/or polluted dust), CALIOP
L3 does not attempt to partition the total extinction into disjoint dust and 'not dust' components. As a result, the CALIOP L3 data used here includes only the 'pure dust' aerosol layer type. In contrast, LIVAS separately extracts the 'pure dust' extinction component from all CALIOP dust aerosol types, including the 'dust,' 'dusty marine,' and 'polluted dust' categories. Consequently, LIVAS can fully describe the spatial and temporal distributions of 'pure dust' in those cases where dust makes only a fractional, perhaps quite small contribution to the total AOD.


Furthermore, the two approaches use different dust lidar ratios (44 sr for CALIOP L3, regionally varying for LIVAS from 40 sr in the Middle East to 56 Sr over the Western Sahara desert region ((Amiridis et al., 2013; Marinou et al., 2017; Proestakis et al., 2018; Floutsi et al., 2022), see supplement). Amiridis et al. (2013) showed that the two products are not similar over the Sahara desert and Europe, with differences occurring due to both different lidar ratios and how different aerosol types are
accounted for.

## 2.4 Conversion of Extinction to Mass Concentration

In order to compare the CALIPSO profiles to those from CAMS, we convert CALIPSO extinction to mass concentration according to Highwood and Ryder (2014),

$$C_{dust} = \frac{1000\sigma_e}{k_e}$$

**Equation 1**

where $C_{dust}$ is the dust mass concentration ($\mu gm^{-3}$), $\sigma_e$ is the extinction coefficient ($km^{-1}$) and $k_e$ is the mass extinction coefficient (MEC, $m^2g^{-1}$), a crucial parameter in linking optical and mass-based measurements. Selection of the MEC is non-



trivial since its value depends on size distribution, particle composition, shape, and may vary in time and space (Ryder et al., 2019; Ryder et al., 2018). Here, to maintain consistency with the comparison to CAMS, we apply the MEC values used

within CAMS for each dust size bin (2.52, 0.94 and 0.41 $m^2g^{-1}$ for size bins 1, 2 and 3 respectively) to the CALIOP extinction. We calculate the vertically-varying proportion of extinction from each size bin in CAMS and then apply this weighting to the extinction in Equation 1 in order to calculate a mass concentration from CALIOP L3 and LIVAS, which thus takes into account a size-weighted MEC for each monthly mean profile. This method assumes that the size distribution represented by CAMS is the same as that sampled by CALIPSO. We choose to compare the profiles using mass, since this is

the metric of interest to the aviation community, though we note that extinction comparisons showed the same results.

## 2.5 Dose Calculations

We calculate the dose of dust ingested by the engine core, imposed by the dusty environment in each location per flight, either for arrival (descent) or departure (ascent). Dust dose is defined as the total mass (g) of dust ingested over a time period (different to the dosage which is the rate of dust ingestion). Dust dose is calculated from (Clarkson, 2020) according to:

$$dose = \int \frac{k_f w_{core} C_{dust}}{\rho_{air}} dt$$

**Equation 2**

where $k_f$ is a dust concentration/dilution factor linked with the engine regime, $w_{core}$ is the mass flow of air entering the engine core (kg $s^{-1}$), $C_{dust}$ is the ambient dust concentration (kg $m^{-3}$, taken from either CAMS or lidar datasets), $\rho_{air}$ is the density of air (at a particular location, altitude and time taken from CAMS) and dt is the time integral, accounting for time spent in

different engine regimes and flight phases (such as taxi, take-off) and at different altitudes. Variation of $w_{core}$, $k_f$ and aircraft altitude is displayed in Figure 2 and given in the supplement.

$w_{core}$ accounts for the operation of the engine impacting the air volume intake via effects on air flow and engine temperature: it is calculated using a mathematical model of the engine performance, matched to real engine data collected from controlled

tests. The mathematical model is based on established theromodynamics and characteristic behaviours of turbomachinery. For example, $w_{core}$ would be lower during descent than take-off due to different engine thrust. $k_f$ results from two effects: larger particles being entrained into the core streamtube, and the centrifuging of larger particles by the engine fan. $k_f$ is influenced by fan blade geometry, flight speed and engine thrust as well as aerosol size distribution, and typically has values of 0.7 to 0.9 for dust or ash (Vogel et al., 2019). We use values of $w_{core}$ and $k_f$ as well as climb and descent rates based on a

modern high bypass ratio turbofan engine provided by Clarkson (2020) and Vogel et al. (2019). Values are representative of a ~70,000 lbf thrust engine, which is appropriate for a long-haul aircraft. Although values will vary for factors such as different take-off weights, day temperatures and how new the aircraft is, the maximum variation in these factors is expected to be under the order of 10%.





Dust dose is calculated at each airport for a single profile ascent or descent from each monthly mean dust profile for CAMS, CALIPSO L3 and LIVAS. Variations in time spent by the aircraft at different altitudes, engine power, dust concentration, $k_f$ and air density contribute to dust dose, as given by Equation 2 and height-resolved values in Figure 2. Dust dose is then averaged temporally to create a dust dose climatology. Dose is calculated separately for departure flight phases (taxi, take-off and climb) and arrival (descent, hold and approach, ground) as well as for the entire flight descent or ascent. The hold phase

assumes a hold altitude of 3,000 ft (~1 km) for 10 minutes. Most holding patterns are in the 10,000 – 15,000 ft (~3 – 5 km) range, though 6,000 ft (~2 km) is not uncommon. We use 1 km for hold pattern altitude here in order to illustrate extreme exposure from dust, and test the sensitivity to hold altitude in Section 4.2.

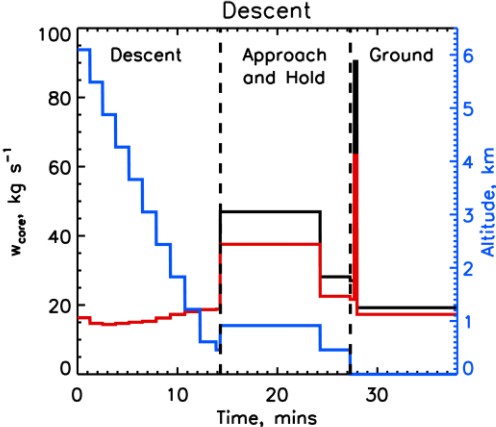
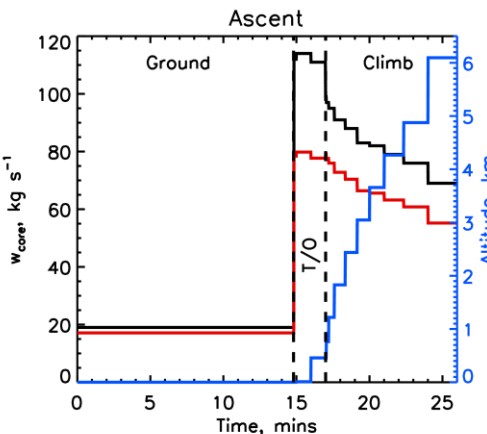

**Figure 2: Variation with time of aircraft altitude, $w_{core}$, and $w_{core}$ multiplied by $k_f$ (red line) for descent and ascent. Text and dashed black lines indicate different flight phases (T/O = take-off).**

## 3   Results

### 3.1  Airport Dust Climatology

Figure 3 shows the seasonal mean dust concentration profiles for each airport using CAMS. Highest concentrations are

generally found in spring and summer, and are particularly notable for Niamey, Dubai, Delhi, Beijing and Marrakesh, with mean values exceeding 25 µg/m³ in at least two seasons. During JJA dust exists in elevated layers, peaking at around 2 km for Niamey, Marrakesh, Beijing and the Canary Islands, reflecting the long-range dust transport in elevated layers occurring with transport from the Sahara desert and Gobi/Taklamakan deserts (Liu et al., 2008b; Yu et al., 2015). This contrasts to Dubai, near to dust sources, where dust peaks closer to the surface at ~200 m, with highest concentrations in JJA. Delhi

shows the highest peak JJA concentration of nearly 175 µg/m³ at around 800 m. All airports show the highest mean dust concentrations in JJA (driven by peak solar heating and dry convection over northern hemisphere desert regions) except Beijing and Niamey. Beijing mean concentrations peak in spring at ~85 µg/m³ at 1.5 km, driven by strong mid-latitude



cyclone surface winds and low precipitation, giving rise to dust transported from the western Chinese deserts towards the Pacific Ocean (Uno et al., 2008; Han et al., 2022). The maximum concentrations in MAM at Niamey are a transition point

when both elevated dust from the Sahara, as well as some contribution from low level dust driven by Harmattan winds contribute to high dust concentrations at both low and mid-altitudes. Sydney, Phoenix, Hong Kong and Bangkok all display mean dust concentrations below 10 µgm⁻³.

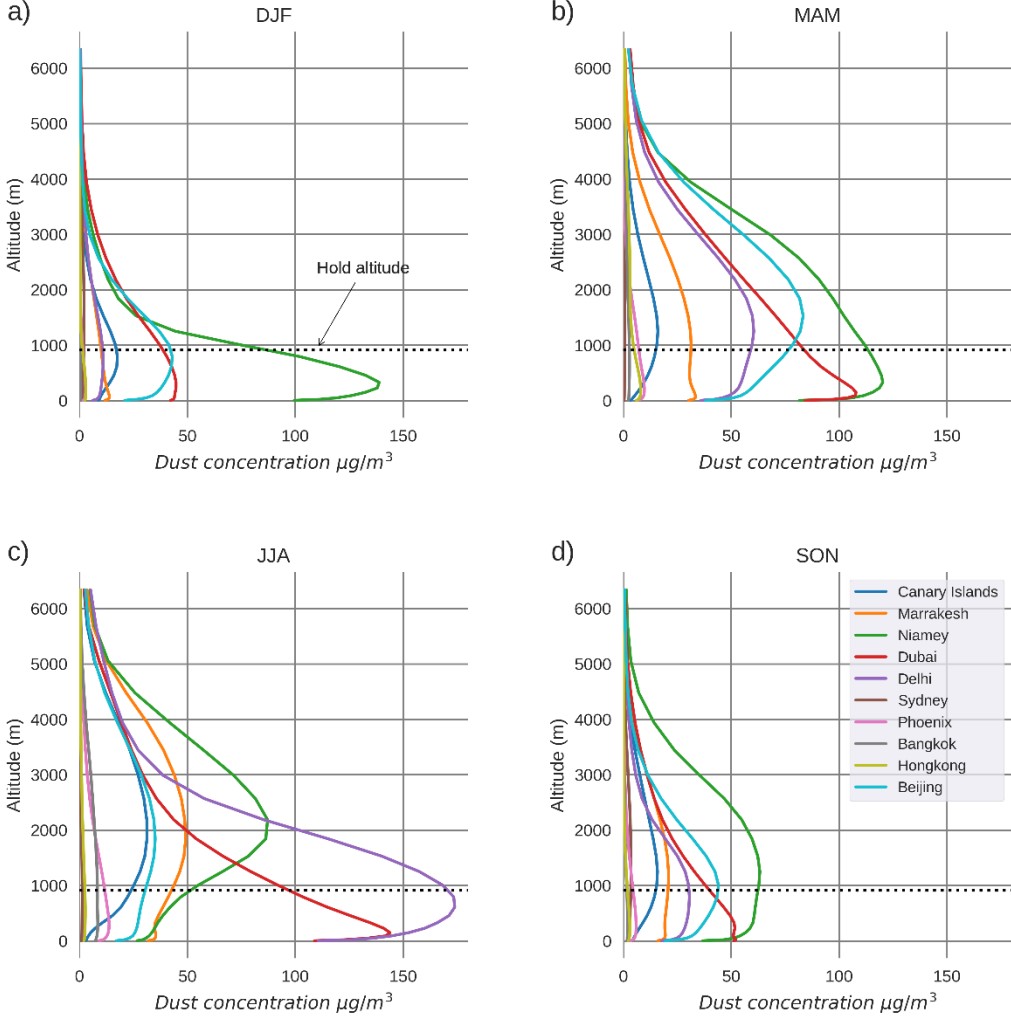

**Figure 3: Seasonal mean dust concentration profiles from CAMS for 10 airports, for 2003-2019, for a) DJF; b) MAM; c) JJA and**
**d) SON.**



Figure 4 and Figure 5 show a comparison between the seasonal mean mass concentration profiles from CAMS, CALIOP L3 and LIVAS at the Canary Islands and Dubai respectively, as well as the factor difference between the lidar retrievals and CAMS. Very good agreement is found between the lidar datasets and CAMS for the Canary Islands (Figure 4), with similar
vertical structure, seasonal cycle and dust concentration magnitudes, though in winter CAMS has the dust peak at a slightly more elevated altitude (1km) compared to the observations which peak closer to the surface.

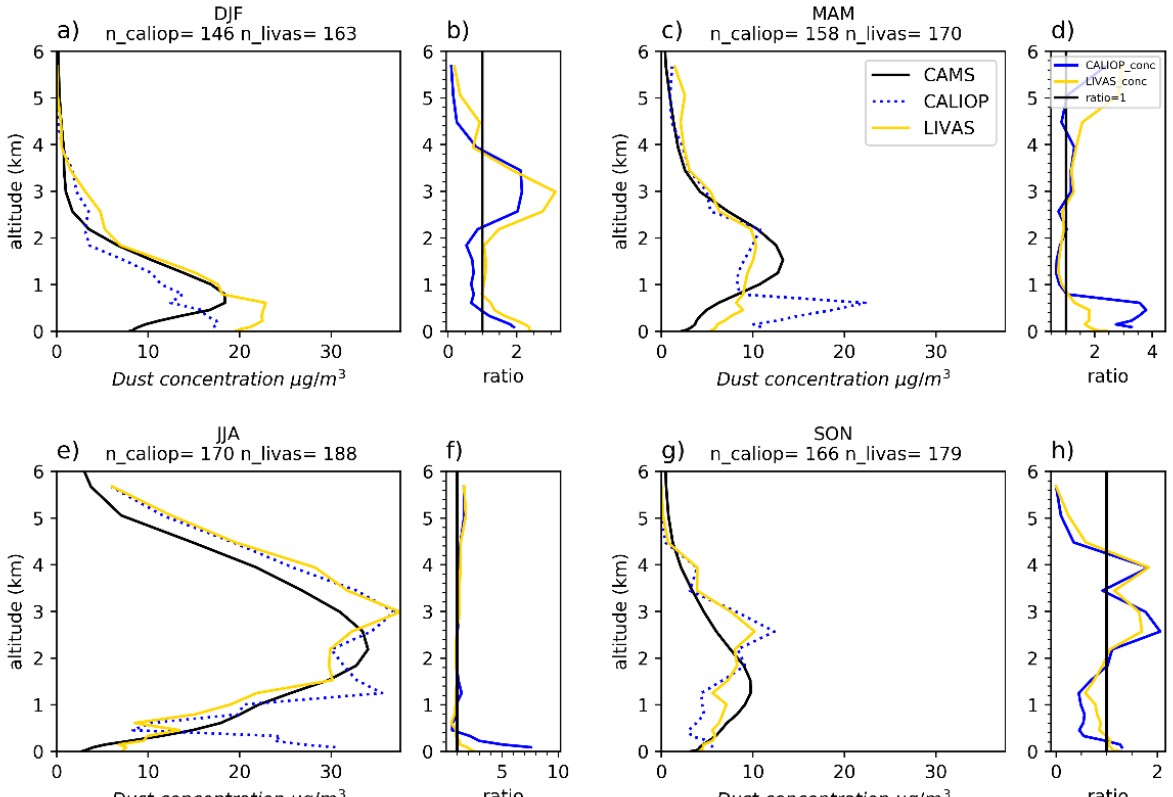

**Figure 4: Canary Islands mean seasonal night-time dust concentration profiles for CAMS (black), CALIOP L3 (blue dots) and**
**LIVAS (yellow) (a, c, e, g). Right hand column sub-figures (b, d, f, h) show the ratio of CALIOP L3 to CAMS (blue) and LIVAS to CAMS (yellow). Note that the scales of each ratio plot differ. Number of CALIOP L3 and LIVAS overpasses is given for each season. CAMS data are only included if coincident with CALIPSO data. All data cover 2007-2019.**



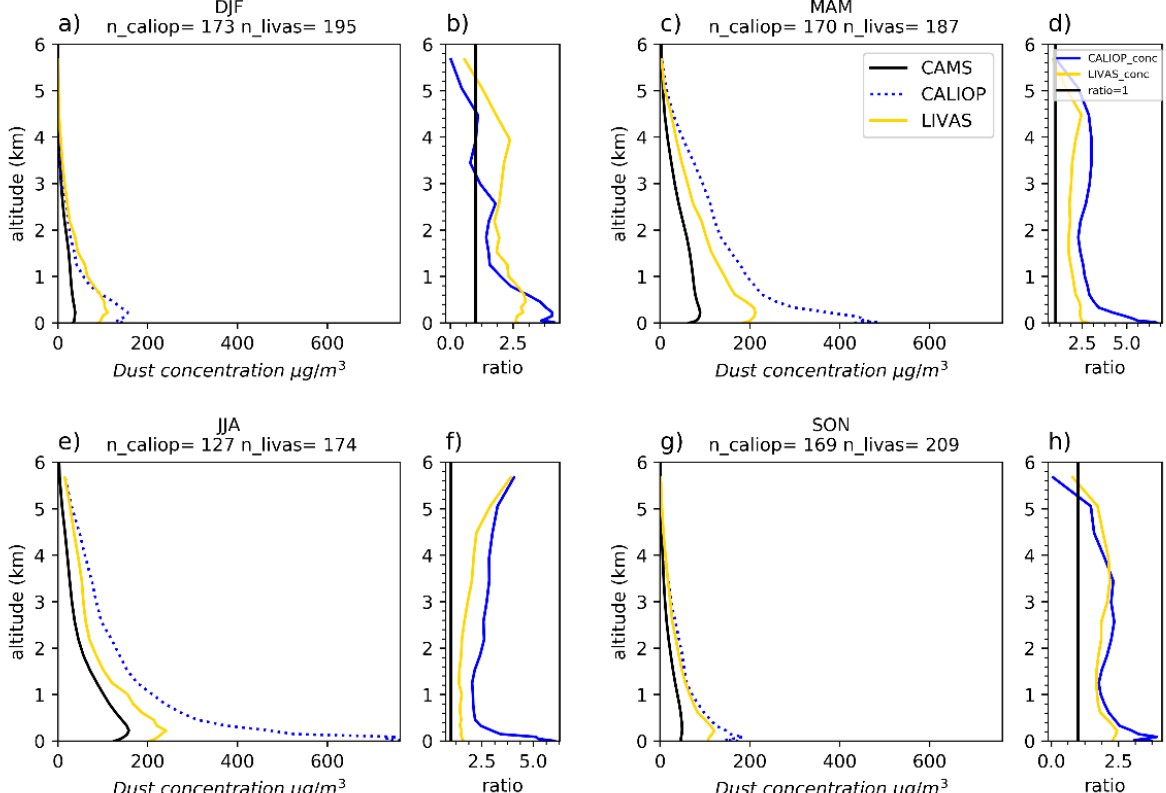

**Figure 5: Same as Figure 4, except for Dubai.**

At Dubai (Figure 5), CALIPSO L3 and LIVAS display the same exponential decrease in dust concentration with altitude as CAMS, and the same seasonal variations with greatest concentrations in spring and summer. However, both lidar datasets are significantly larger in magnitude than CAMS throughout the year, with CALIOP L3 exceeding a factor of 5 greater than

CAMS at the lowest altitudes. LIVAS displays lower dust concentrations than CALIOP L3, but is still larger than CAMS by up to a factor of 2.5 in the lowest 1 km. Between 1 km and 5 km both lidar datasets are within a factor of 2.5 of CAMS.

Similar comparisons were performed for the other airports with significant dust loadings (see supplement). Similar to Dubai, at Niamey, Marrakesh and Delhi the shape of the vertical profile and seasonal cycles agree, though the lidar magnitudes are a

factor of 2-3 greater than CAMS for Niamey and Marrakesh, but are comparable for Delhi in JJA, the peak dust season. At Beijing, although the seasonal cycles were similar and dust concentrations on the same order of magnitude for CAMS and the lidar datasets, the vertical structure is very different with CAMS showing an elevated peak at around 2km while the observations showed an increase towards the surface. Generally LIVAS concentrations are substantially lower than CALIOP





L3, and therefore closer to concentrations from CAMS. However, in Beijing and Marrakesh dust concentrations from
LIVAS are greater than CALIOP L3.

## 3.2 Engine core dust dose

### 3.2.1 CAMS

Figure 6 shows the climatological engine core dust dose calculated from CAMS for the six dustiest airports for aircraft
arrival descent. Total dose (the sum of approach, hold, descent and ground doses) is highest for Delhi in JJA, at 6.6 g. Total
dose varies by airport and season, with the highest values overall being found for Niamey, Delhi and Dubai. Delhi and Dubai
encounter the greatest doses in JJA (6.6 g and 4.3 g), while Beijing and Niamey are greatest in MAM (2.9 g and 4.7 g),
reflecting the seasonal changes in dust concentration profiles (Figure 3). Niamey stands out as the only airport with dose
greater than 2 g in winter.

The greatest contribution to dose for the flight profile under consideration comes from the 'approach and hold' phase, which
contributes at least 50% of total dose in all cases. This is because the hold altitude of around 1 km frequently coincides with
the altitude of peak dust concentration (Figure 3), or is very close to it. Nevertheless, the dose magnitude from approach and
hold is driven by seasonal and regional variations in dust concentration. We note that hold altitude of 1 km was selected to
illustrate extreme exposure that can result from low holding altitudes, and that hold patterns will not always dominate
descent dust dose (Section 4.2).

The relative contributions from descent and ground phases to total arrival dose varies seasonally. For example, at Niamey
there is a much greater contribution from descent to total dose in JJA when the dust plume is elevated, rather than in DJF
when the contribution from the ground phase is greater due to the plume being close to the surface.








**Figure 6: Climatological aircraft arrival engine core dust dose for each airport and season, separated by different flight phases and total dose. Data from CAMS covering 2003-2019.**

For departure (Figure 7), generally total dose is slightly lower than for arrival (24% lower on average over all airports). This is due to the smaller amount of time spent departing (26 vs 38 minutes), despite significantly higher $w_{core}$ values during departure, indicating higher engine power, as well as the lack of a holding phase which significantly contributes to arrival dose. The highest departure dose reaches a value of 4.4 g for Delhi in JJA. Similar to arrival, departure dust dose for all airports is highest in MAM and JJA, reflecting the seasonal changes in dust profiles.



**Figure 7: Climatological aircraft departure engine core dust dose for each airport and season, separated by different flight phases and total dose. Data from CAMS covering 2003-2019.**

The flight phase contributing to the greatest proportion of total departure dose varies by season and airport. For airports with an elevated dust plume during MAM and JJA, such as Niamey, Beijing, Marrakesh, and the Canary Islands, the climb phase contributes the greatest proportion to total dose. Contrastingly, for airports such as Dubai and Delhi, where dust concentrations peak closer to the surface, all three flight phases contribute roughly equally to total dose. Despite the small amount of time (just over 2 minutes) spent in the take-off phase, take-off often constitutes a disproportionate second largest

contribution to total dose, due to a combination of extremely high values of $w_{core}$ and frequent coincidence with maximum or high dust concentrations.





### 3.2.2 CAMS vs spaceborne lidar calculated dose

Next we compare the dust dose calculated from CAMS to those calculated from CALIOP L3 and LIVAS data: firstly in terms of magnitude, then in terms of seasonal cycle. Figure 8 shows the magnitude of arrival dust dose calculated from

CAMS to those calculated from CALIOP L3 and LIVAS data. For certain airports such as the Canary Islands, Beijing, and to a certain extent Marrakesh, the agreement between CAMS and the lidar datasets is good - for these airports the CAMS median resides within the interquartile range of both lidar datasets.

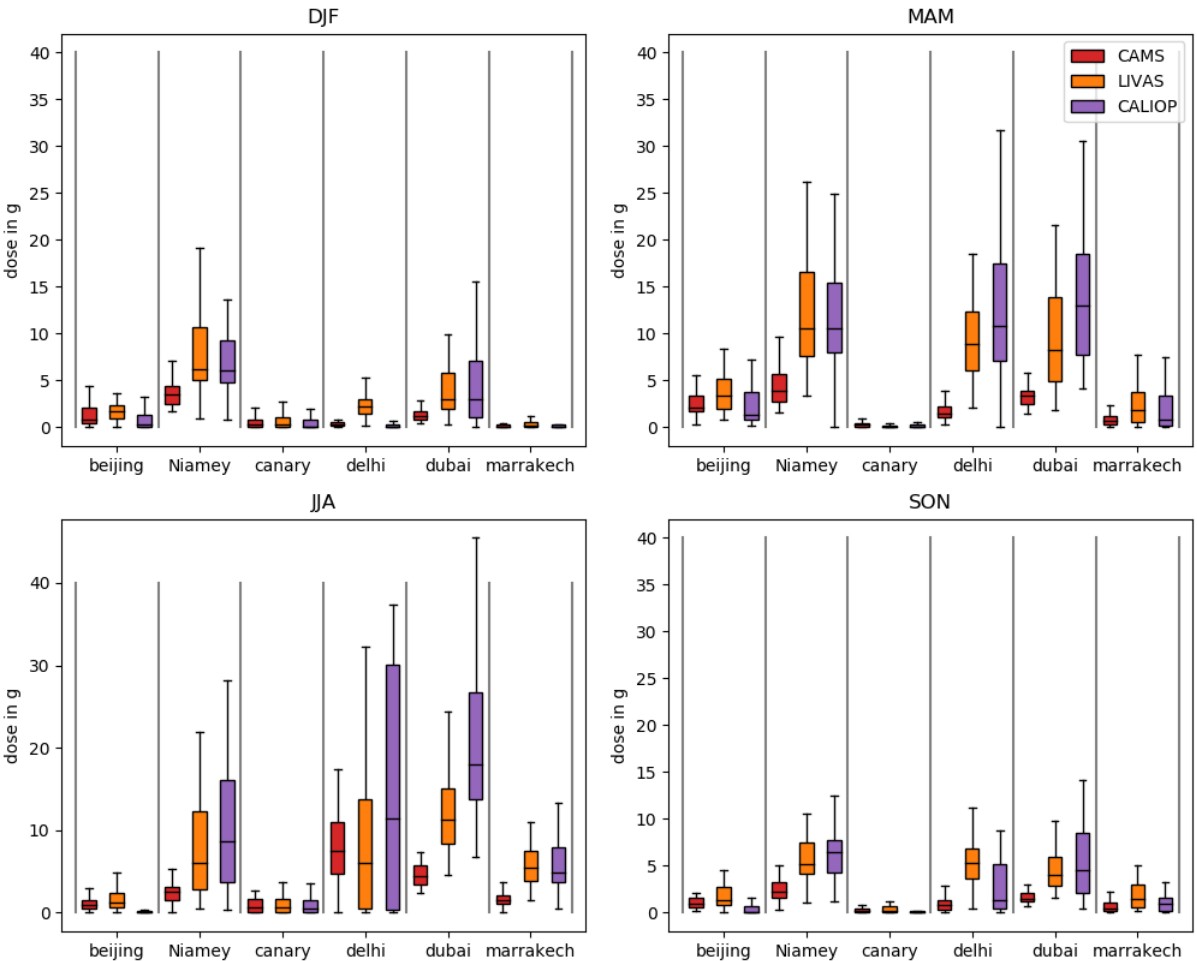

**Figure 8: Seasonal engine core dust dose calculated for each airport and season for arrival, for CAMS, LIVAS and CALIOP L3 datasets. Box plots show median and interquartile range, whiskers show an additional factor of 1.5 of the interquartile range. All data cover 2007-2019.**

The largest differences, with LIVAS values larger than CAMS, and CALIOP L3 substantially larger than both, are most

evident at airports with a low altitude dust plume, particularly Niamey in DJF/MAM and Dubai year-round. This is because



the different datasets give very different dust concentrations close to the surface, which contributes to very different doses from the ground and hold phases, making up the largest part of descent dust dose. Even when the Niamey plume is elevated in JJA the doses between datasets are different (CALIOP L3 and LIVAS are a factor of 3.4 and 2.4 larger than CAMS), since the calculated dose is strongly dependent on the exact dust concentrations at the 1km holding altitude which varies between

datasets, and CAMS dust concentrations are underestimated compared to the lidars at all altitudes. For Dubai, agreement is better between CAMS and LIVAS than CAMS and CALIOP L3, with CALIOP L3 median dose being 3.4 times larger than CAMS, compared to a factor of 2.6 for LIVAS. In Niamey the lidar datasets give a dose 1.7 to 3.4 times that from CAMS, though agreement between CALIOP L3 and LIVAS is much better, predominantly because dust concentrations happen to be similar at holding altitude (1 km) and at the surface, even though CALIOP L3 concentrations are around 50% larger than

LIVAS in between these altitudes.

In Delhi the LIVAS dust dose is higher than both CAMS (by 6-7 times) and CALIOP L3 in DJF and SON. This is due to the dust concentration profile being larger for LIVAS than CALIOP L3 at nearly all altitudes in these seasons (see supplement). In MAM CAMS shows a much lower dose than LIVAS and CALIOP L3, while in JJA median CAMS dust doses are similar

to LIVAS and CALIOP L3, though CALIOP L3 shows a larger spread.

Beijing is a second airport where the dose from LIVAS is the largest of the three datasets, a factor of 1.6 larger than CAMS. The dust concentration profiles show LIVAS to be greater than CALIOP L3 across most altitudes in all seasons, and while LIVAS shows similar magnitudes to CAMS, LIVAS profiles show a stronger surface concentration with declining

concentrations in altitude, while CAMS shows a smoother profile with more elevated peaks. However, although the vertical profiles are very different, the dose calculations are fairly similar since concentrations around the holding altitude are similar, and to some extent the differences in vertical distribution compensate. CALIOP L3 shows lower doses for Beijing across all seasons, by a factor of 0.3, as a result of the smaller dust concentration profiles.

Overall there are large variations in magnitude of dose at some airports, particularly those featuring large surface concentrations and those where concentrations at holding altitude vary between the different datasets. Possible reasons behind the differences are explored in Section .

For departure (see supplement), overall the similarities and differences between the datasets are the same as for arrival, with

lower doses for departure by 10 to 23%. However, in a few cases differences between datasets compared to arrival are sensitive to the overall vertical profile shape and magnitude, particularly if ground concentrations are very large. If concentrations are high near the ground, the extra engine power utilized during take-off, as well as the additional time spent on the ground in the taxi phase, can result in higher departure dose compared to arrival dose. For example, for Dubai in JJA, the total dose from CALIOP L3 is 18% higher for departure than arrival. This is because the departure dose is particularly



sensitive to high dust concentrations at very low altitudes, and the CALIOP L3 dust concentrations in the lowest layer are nearly 4 times as large as CAMS/LIVAS. Similarly, for Beijing in MAM, greater low altitude dust concentrations from LIVAS result in a greater increase in dose from LIVAS compared to CAMS/CALIOP L3 for departure, rather than the compensation between different profile altitudes which occurs for arrival.

Other notable features are that the variability is much larger for the lidar datasets compared to CAMS, with CALIOP L3 showing slightly more variability than LIVAS. This is partly a feature of the larger magnitudes seen in the lidar data compared to CAMS, but also due to the lower sampling rate for CALIPSO compared to the regular 3 hourly model output from CAMS. Additionally, there may be differences between CALIOP L3 and LIVAS in filtering the largest, but less frequent, dust events. Section 5.2 explores this further.


Figure 9 shows the normalized seasonal cycles of dust dose in order to evaluate the seasonal timing of dust dose variability. CAMS replicates the seasonal cycles seen in the lidar data very well at Dubai, Canary Islands and Marrakech, but less well at Niamey, Beijing and Delhi. At Niamey CAMS underestimates the relative summer dust dose and overestimates the winter dose. At Beijing there is a suggestion that CAMS represents the spring peak dust dose too broadly, though the lidar datasets

display disparities too. At Delhi, CAMS offsets the peak dust dose towards the late summer compared to the lidar datasets, leading to substantial differences in May and July, and also misses the secondary dust dose peak in autumn. Overall CAMS represents most of the broad seasonal variability well, but struggles to capture more detail at certain airports.



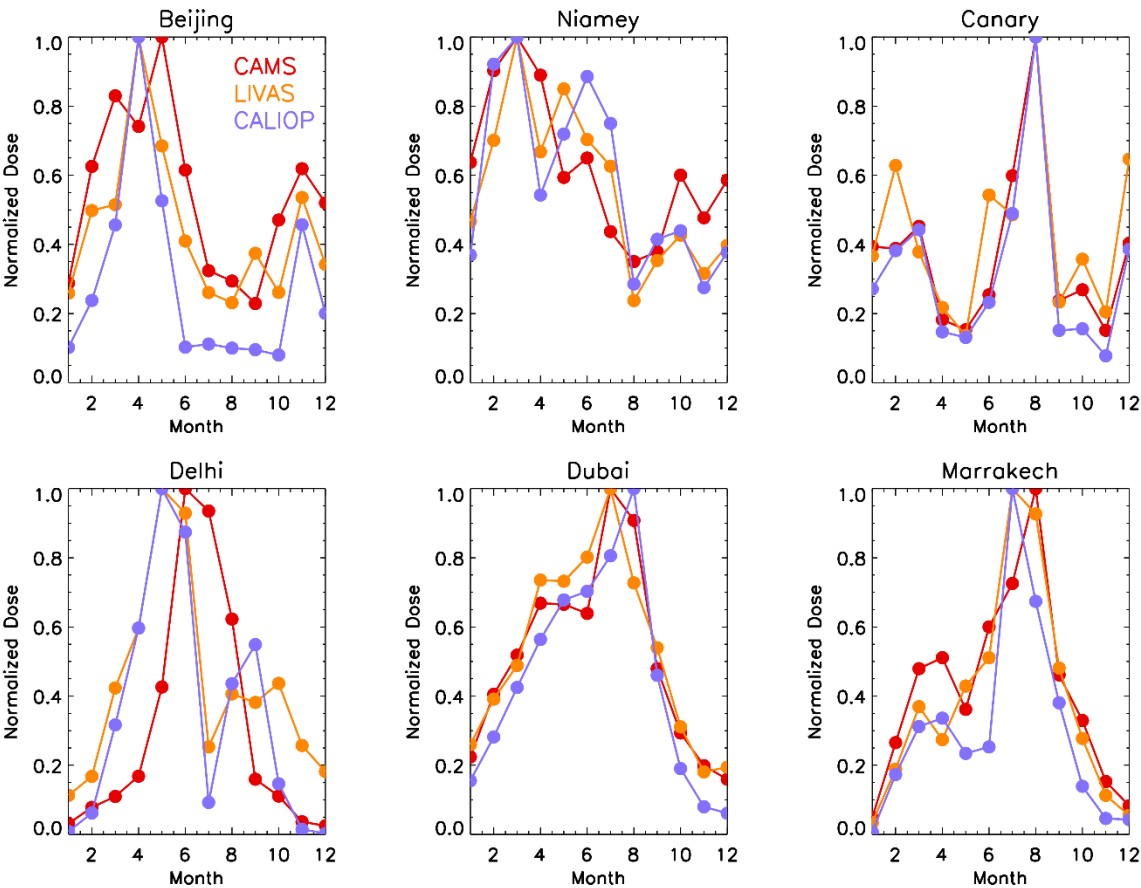

**Figure 9: Seasonal variation in normalized arrival dust dose for each airport and dataset. Dose at each airport and for each dataset is normalized by the month with the largest dust dose.**

## 4    Measures to reduce dust dose

### 4.1  Diurnal cycle in dose

Airports where dust is relatively local and subject to intense solar heating are subject to strong diurnal cycles in the dust concentration profile (Cuesta et al., 2009; Kocha et al., 2013). Figure 10 illustrates the CAMS mean seasonal changes in dust concentration profile at Dubai for the maximum (1600-1900 local time) during the day and the minimum during night (0100-0400 local time). During all seasons there is a strong diurnal cycle in the dust concentration, and the diurnal cycle is strongest during spring and summer. During the day, downward mixing of the nocturnal low level jet followed by intense solar heating drives local convection and dust uplift. At night time, some of this dust remains in a slightly elevated layer while surface concentrations drop due to reduced turbulent mixing of dust from the surface. This is a well-studied



phenomena impacting dust in arid regions. However, it is notable for engine ingestion purposes that night near-surface concentrations can drop to near to half of their peak day time values.

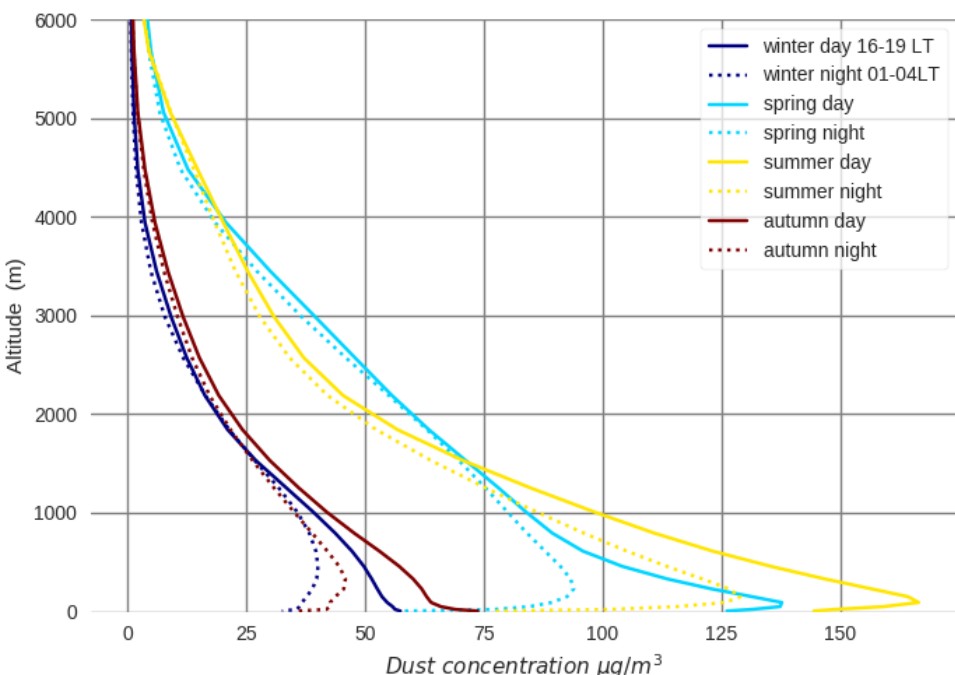

**Figure 10: Diurnal cycle of dust concentration at Dubai from CAMS. Solid lines indicate daytime, dotted lines indicate night time. Day times are selected as 1600 to 1900 local time, night times as 0100 to 0400 local time.**

In Table 1 we show the reduction in dust dose possible for each airport between the maximum and minimum throughout the diurnal cycle, for an arrival immediately followed by a departure, equivalent to aircraft delaying arrival and departure from late afternoon to night time. Since dust diurnal cycles are largest for profiles with high near-surface concentrations, reductions in dose due to night time flying are largest for airports with these characteristics: Dubai in JJA (5.95 g reduction), Delhi in JJA (6.44 g) and Niamey in DJF (5.00 g). For the peak dust seasons, a reduction in dose of 41% at Dubai in JJA, 34% at Delhi in JJA and 39% at Niamey in DJF could be achieved. Lower relative reductions are found at airports with a an elevated dust plume, such as the Canary Islands (5-14% reduction), Marrakesh (17-22% reduction) and Niamey (20% in JJA when the plume is elevated). Moderate reductions are found at Beijing (29-33 %). Substantial reductions in total dose are possible by adjusting time of day for arrival and departure.





| Airport | DJF | | MAM | | JJA | | SON | |
|---|---|---|---|---|---|---|---|---|
| | Dose reduction (g) | % reduction | Dose reduction (g) | % reduction | Dose reduction (g) | % reduction | Dose reduction (g) | % reduction |
| **Beijing** | 1.23 | 33 | 2.19 | 29 | 1.03 | 29 | 1.28 | 33 |
| **Niamey** | 5.00 | 39 | 3.85 | 28 | 1.23 | 20 | 1.74 | 26 |
| **Canary Islands** | 0.19 | 14 | 0.18 | 17 | 0.38 | 19 | 0.05 | 5 |
| **Delhi** | 0.23 | 22 | 1.67 | 27 | 6.44 | 34 | 0.91 | 29 |
| **Dubai** | 1.84 | 34 | 4.94 | 39 | 5.95 | 41 | 3.08 | 44 |
| **Marrakesh** | 0.31 | 22 | 0.68 | 17 | 1.19 | 21 | 0.46 | 19 |

**Table 1: Seasonal mean reduction in dust dose (g) between maximum and minimum throughout the diurnal cycle for an arrival directly followed by a departure for the CAMS dataset. Final column indicates the percentage reduction in dust dose possible for each airport's peak dust season.**

## 4.2 Mitigation via holding altitude

The holding phase accounts for the largest proportion of dose during arrival. Therefore Figure 11 shows how selecting a different hold altitude can impact dust dose, compared to a hold altitude of 1000 m. In general dose decreases as holding altitude is raised, because for most airports dust concentration decreases with height above 1000 m. For airports such as Delhi and Dubai in JJA the changes are particularly striking, since peak dust concentrations coincide with holding altitude (Figure 3) and dust concentration drops rapidly above this altitude. Here dose from hold can be reduced by 75% (Delhi) and 63% (Dubai) by increasing hold altitude from 1 km to 3 km, equivalent to a reduction in total descent dose of 41% and 31% respectively. During spring all airport profiles exhibit broader profiles, so an increase of holding altitude to 4 km would be necessary to see substantial reductions from holding dose. Conversely, in DJF dust concentration drops off much more rapidly due to reduced boundary layer depth, providing large relative reductions from hold at 2 km.

A different pattern is shown for certain airports where an elevated dust plume is present in certain seasons, notably for Niamey in JJA where the hold dose actually increases by around 30% when holding altitude is raised to 2 km. This is also the case for the Canary Islands, Marrakesh and Beijing in JJA, though total changes in dose are much smaller (around 0.02-0.04 g), though the relative increase in hold dose can still be up to 30%. However, the vertical distribution of dust concentration from CAMS is significantly different to the lidar datasets for Beijing in MAM, JJA and SON, with CAMS indicating a dust peak at around 2km, while the lidar datasets indicate increasing concentrations towards the surface (see



supplement). This will have implications for selecting a hold altitude with the least dust dose, and a choice based on the
CAMS reanalysis dataset may be inaccurate.

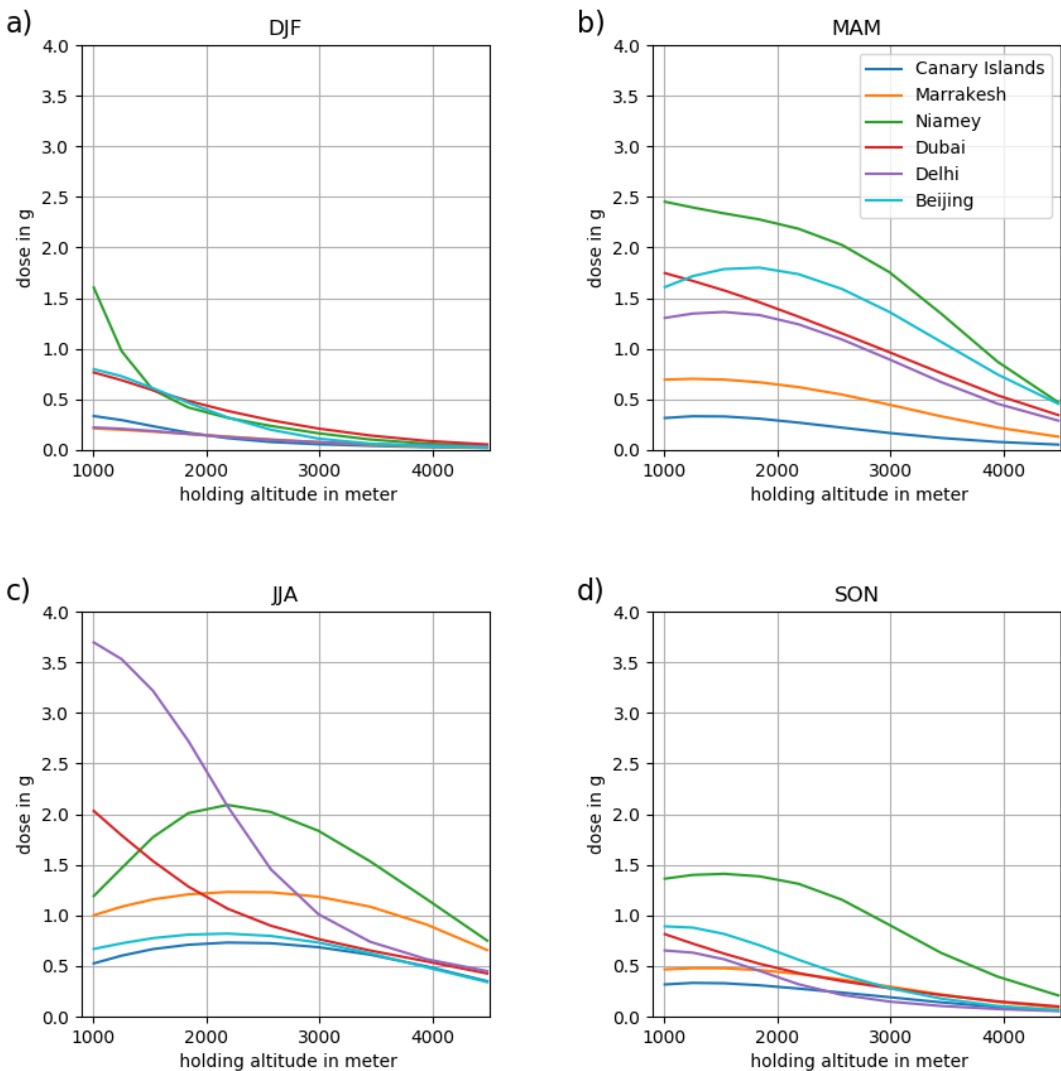

**Figure 11: Dust dose (from holding pattern of 10 minutes) dependence on holding altitude from CAMS for each airport and
season.**

**5    Discussion**

Here we explore some of the causes and possible explanations for differences between the three datasets.



## 5.1 CAMS vs Lidar datasets

1. The lidar profiles are filtered for clear sky (i.e., cloud-free) and have much larger grid squares over which they are averaged (2 by 5 degrees, in order to improve sampling statistics) compared to CAMS (80 km). Therefore the differences in temporal and spatial sampling introduce unquantified differences between the datasets and contribute to the differences in smoothness between CAMS and the lidar datasets.

2. Likely uncertainty in CAMS originates from the data assimilation (DA) of satellite-derived aerosol optical depth (AOD) into incorrect aerosol speciations and/or vertical distribution. For example, if the first model guess contained too little dust compared to other aerosol species, and AOD was then incremented as part of the DA process, the dust optical depth (DOD) (and therefore dust mass concentrations) would remain too low despite the assimilation of satellite aerosol retrievals. The sulfate aerosol species receives the most increments on a global scale from DA due to its ubiquity and relatively long lifetime. This may explain the reason why CAMS often shows too little dust compared to the lidar datasets, particularly closest to the surface, since at these altitudes total aerosol is more likely to exist as a mixture of different aerosol types. In locations more likely to contain pure dust without mixing from other aerosol types (e.g., Marrakesh and the Canary Islands), agreement between CAMS and the lidar datasets is much better.

3. Although CAMS has been shown to represent annual, spatial and seasonal variability very well for total aerosol, with good agreement with observations (Errera et al., 2021), CAMS reanalysis DOD is known to be somewhat lower than found in other datasets, such as MERRA-2 (e.g., Zhao et al. (2022)). The low bias of DOD over dusty regions is much more pronounced than that of total AOD, because of the issue of wrong speciation of data assimilation increments mentioned under point 2 above. Therefore we would expect comparisons against extinction or mass from lidars to be similarly low. CAMS reanalysis DOD underestimates have been shown to be on the order of around 40% compared to AERonet RObotic NETwork (AERONET) observations (Errera et al., 2021), which would roughly explain the factor of 2 or greater differences seen in this study.

4. In general, CALIOP total AOD estimates are biased low relative to other instruments (e.g. Sayer et al. (2018); Schuster et al. (2012); Song et al. (2021). This is because CALIOP only retrieves extinction coefficients in those regions of a profile where aerosols have been detected, whereas passive sensors that cannot make height-resolved measurements necessarily retrieve optical depth estimates for the entire atmospheric column. Specific to dust, Schuster et al. (2012) found a -29% bias in CALIOP AODs in an exclusive dust environment compared to AERONET, while Song et al. (2021) found global annual mean DAOD less than a factor of two smaller than those from MODIS. Therefore it is surprising that despite this, the lidar mass profiles and doses tend to be higher than CAMS.

5. A significant assumption is the values applied for the MEC in converting between extinction and mass concentration (optical and mass parameters). If the dust composition or size distribution sampled by CALIPSO



differs from that assumed or modelled in CAMS, then the conversion will be imperfect. Given a fixed composition, MEC is larger for smaller dust particles (Ryder et al., 2013; Ryder et al., 2019). Therefore in order to bring CAMS and CALIOP into better agreement, larger MEC values (representative of a greater proportion of smaller particles) would be needed to be applied to the lidar datasets. However, it has been shown that in Saharan dust plumes, CAMS may overpredict fine particles while underpredicting the coarse ones (O'sullivan et al., 2020), a common

feature in dust models (Adebiyi and Kok, 2020; Adebiyi et al., 2023). The largest CAMS size bin covers a wide range of diameters (1.8 to 40 µm) with a single value of MEC. If the lowest dusty layers of the atmosphere are dominated by particles in this size range, but are more biased towards the lower end of the size range, this would justify a larger MEC at these altitudes, which would narrow the gap between the datasets.

6.     CALIOP uncertainties increase nonlinearly with increasing penetration through a layer, more so where particulate

concentrations are higher and for high lidar ratios (such as for dust). This frequently leads to higher uncertainties in cases where a thick dust layer is found close to the surface (e.g., Dubai year round, Niamey in DJF and MAM, Delhi in MAM, Canary Islands in DJF) (Young et al., 2013; Yu et al., 2021). High uncertainties at lower altitudes can indicate either overestimates or underestimates of the corresponding extinction coefficient. However, for the most part they tend to flag large overestimates, which may explain why airports with high surface dust

concentrations show notably lower dust mass concentrations for CAMS compared to the lidar datasets.

Overall it seems most plausible that underestimates of CAMS dust mass concentrations due to the challenge of aerosol increments by DA, described in points 1-3 above are the cause of the differences between CAMS and the lidar datasets.

## 5.2 CALIOP L3 vs LIVAS

1.     Both lidar datasets are dependent on a choice of lidar ratio to convert measured attenuated backscatter signal into extinction coefficient (44 Sr for CALIOP L3, regionally varying for LIVAS). In reality the lidar ratio will vary spatially with dust optical properties (Schuster et al., 2012; Floutsi et al., 2022). This contributes to differences between LIVAS and CALIOP L3. For example, for Saharan dust regions where the LIVAS lidar ratio (53-56 sr) is larger than the CALIOP L3 lidar ratio it may explain the Marrakesh dust concentrations being larger in LIVAS, and

possibly also Niamey. The slightly lower Middle East lidar ratio of 40 sr in LIVAS may contribute to the lower mass concentration profile in Dubai, though cannot feasibly explain the factor of 5 differences in some seasons. For the Asian airports (Beijing and Delhi) with a LIVAS lidar ratio of 46 sr, the small difference in lidar ratio compared to CALIOP L3 cannot explain the large differences in dust concentration. Therefore lidar ratio differences are not likely to be the dominant cause of differences between CALIOP L3 and LIVAS in this study.

2.     CALIOP L3 and LIVAS use different depolarization thresholds for discriminating between dust and other aerosol types. CALIOP L3 uses a threshold of 0.20 while LIVAS uses 0.31. However, LIVAS also uses this threshold to extract the dusty volume from air containing a mixture of dust and other aerosol types with depolarization values





between 0.05 and 0.31, which is then included in the LIVAS dust data. We did not include the 'polluted dust' or 'dusty marine' categories from CALIOP L3 since they incorporate other aerosol types besides dust, while LIVAS

includes these classifications but extracts the portion of extinction within them derived from dust. Therefore, the LIVAS data includes the contribution from dust in mixed aerosol scenes while CALIOP L3 does not. These different selection approaches very likely result in different climatological profiles depending on the relative proportions of dust occurring in mixed aerosol scenes compared to the dust-only cases:

a. When mixed aerosol type conditions occur in lower concentrations than pure dust cases, the inclusion of

the dust profiles from the mixed cases in the LIVAS data will lower the mean LIVAS extinction profile compared to CALIOP L3. This occurs at low altitudes at Dubai year round, Niamey in DJF and MAM and Delhi in MAM.

b. When mixed aerosol type conditions occur in higher concentrations than typical pure dust cases for a location, the opposite occurs, and exclusion of the dust from the mixed aerosol profiles in the CALIOP L3

data will lower the mean CALIOP L3 profile compared to LIVAS. This occurs in Beijing in all seasons and is clearest near the surface, and is also the case for Delhi in DJF and SON.

This difference in selection and filtering of profiles most likely explains the main differences between LIVAS and CALIOP L3 found in this study. Inclusion of the mixed dust profiles from CALIOP L3 could be applied to test this hypothesis. We cannot explicitly confirm this from the analysis performed, and further exploration is beyond

the scope of this article.

3. Differences are also found in the very lowest altitudes (the lowest 2 data points) between CALIOP L3 and LIVAS in cases where a very dusty profile peaks close to the surface. In these cases, CALIOP L3 concentrations drop rapidly in the lowest 2 altitude bins while LIVAS do not (Niamey DJF and MAM; Delhi JJA and MAM). This is likely to be because the lidar signal is attenuated before reaching the lowest dust layers and the dust layer base is

assigned to 90 m altitude (Section 2.3.1). These cases are then excluded from CALIOP L3 data (Tackett et al., 2018). Contrastingly, we set the LIVAS data to that of the lowest altitude above where data is absent (Section 2.3.2) such that the dust concentration at these locations does not decrease rapidly towards the surface. This may result in the rapid drop-off of dust just above the surface in certain cases for CALIOP L3, though this explanation would require further investigation. Due to the importance of surface dust concentrations in determining total

dose, this results in the larger CALIOP L3 dust concentrations in the overall profile at these locations being offset by the smaller CALIOP L3 surface concentrations, resulting in similar dust doses between CALIOP L3 and LIVAS in these cases.

Finally, there are inevitably uncertainties in the quantities used to calculate $w_{core}$ in the dust dose calculations. These mainly stem from the difference between the calibrated representative data and actual specific in-service engine and flight details.

For example, the generic data assumes a particular day temperature, humidity, airport altitude, engine condition (usually





brand new) and take-off weight, all of which will vary. The largest variability in the data presented concerns whether a holding pattern was required in a flight, and the chosen altitude for the holding pattern, as explored in Section 4.2.

## 6   Conclusion

Atmospheric mineral dust represents a problem to aviation. There are safety considerations if dust concentrations at airports
are sufficiently high that visibility drops below a few hundred metres, but this concerns the risk of collisions when manoeuvring aircraft on the ground - it is not a concern once aircraft have taken off. With regard to the direct impact dust has on aircraft, the greatest problem is the long term damage to engines from ingesting dust on the ground and flying through dusty regions which degrades engine performance and affects maintenance schedules and resource planning. This study quantifies the climatological vertical distribution of mineral dust at a variety of worldwide airports from modelled and
observed dust concentration data in order to estimate the resulting climatological core engine dust dose for the first time. We take into account typical aircraft ascent and descent rates as well as air and dust engine ingestion rates which vary with altitude, location and engine power, for a representative modern turbofan engine with a rated take-off thrust of approximately 70,000 lbf.

Using the ECMWF CAMS reanalysis, we find that airport dust profiles vary seasonally and regionally as expected, and that broadly variations in engine dust dose reflect these variations. Climatologically, Sydney, Phoenix, Bangkok and Hong Kong were not very dusty, and the nature of dust damage at these airports is only likely to be occasional.

We find dust dose to be largest in JJA followed by MAM for most airports, though some airports experience different
seasonal cycles: Beijing and Niamey suffered their largest dust doses in MAM, as a result of seasonal meteorological features driving dust uplift and transport. The largest arrival dose was calculated for Delhi in JJA (6.6 g), followed by Niamey in MAM (4.7 g) and Dubai in JJA (4.3 g).

It is worth discussing the implications of a jet engine core ingesting around 5 g of mineral dust. Such a small amount of dust,
on its own, represents a negligible problem for a modern jet engine core. However, when it is considered that many engines see this level of dust ingestion every flight (or 'engine cycle') – especially those operated by airlines with hubs of dusty airports – the cumulative dose starts to represent more of a problem; 1000 landings and subsequent departures from a dusty airport represents around 10 kg of cumulative dust ingestion. The impacts of a controlled dust ingestion test, conducted in 2018, which delivered approximately 5 kg of dust with composition representative of region around Dubai, are reported in
Elms et al. (2021). The level of contamination in the engine hot sections was enough to result in accelerated engine performance deterioration and substantially reduced component lives. Therefore the likely level of engine deterioration



experienced by aircraft operating out of dusty hubs represents a significant cost to the aviation industry. Flights operating between two dusty locations, such as Delhi and Dubai, would result in double the dust ingestion.

Further, the specific type of damage caused is a function of when during the engine cycle dust is ingested. Some damage mechanisms only manifest themselves when dust is ingested at high power, such as take-off and climb. Others are driven predominantly by low power ingestion, such as idle during ground operation or during descent. Consequently, knowing how much dust is ingested during each flight phase – as quantified here – is as important as a knowledge of the total dust ingested for a complete flight.


In this study, engine dose from departure was 24% lower than that from arrival due to the large contribution to dose from the holding pattern altitude. During arrival, the hold pattern contributed over 50% of total dose due to the long time (10 minutes) spent in the hold pattern, which was compounded by hold altitude (1km) frequently occurring at, or near to, maximum dust concentrations. Contribution to dust dose from the different flight phases during departure (ground, take-off, climb) was

more varied between airport and dust regime than arrival. For example, airports with seasonal elevated dust plumes (such as Beijing and Niamey in MAM and JJA and the Canary Islands and Marrakesh in JJA) experienced a dominant contribution from the higher altitude climb phase, while airports with seasonal near-surface dust plumes (e.g., Dubai year-round, Delhi in JJA) had similar contributions from all three departure flight phases.

Engine dust dose calculated from the CAMS reanalysis was compared to that from two datasets from the CALIPSO satellite with a spaceborne lidar: the standard CALIOP L3 dataset and the LIVAS dataset. Mostly the seasonal cycles in dose were very similar between CAMS and the lidar datasets, except for minor differences in the seasonal cycle at Beijing and Delhi. Likewise, CAMS mostly represented the shape of the vertical dust structure well, though CAMS tended to be unable to represent strong peaks in dust concentration near the surface seen in certain seasons at Dubai, Delhi and Niamey, and CAMS

failed to represent the vertical structure shape at Beijing.

In terms of magnitude of dust concentration and also dust dose, values were frequently very different between all three datasets. At airports where dust was concentrated close to the surface such as Dubai, CALIOP L3 dust mass concentrations were up to 6 times greater than CAMS, and LIVAS doses up to 2.5 times greater than CAMS. Large differences in dust

concentrations at low altitudes are magnified when calculating dust dose by the large contribution to total dose from the ground and hold phases. In other cases, dose calculated from LIVAS was greater than CAMS and CALIOP L3, such as at Beijing. In 46% of all seasons and airports, CAMS substantially underestimates dust dose compared to both CALIOP L3 and LIVAS, with these datasets larger by a factor of 1.9 and 2.8 respectively, resulting in a mean underestimate by CAMS of 2.4 over both datasets.




The differences in dust dose between CAMS, CALIOP L3 and LIVAS can be traced back to differences in the vertical profiles for each dataset. The comparisons are complex, firstly due to requirements for uncertain and variable properties of dust such as the mass extinction coefficient and its contribution from different dust size ranges in converting between mass-based and optical quantities, secondly due different thresholds and methods for identifying dust in the lidar datasets, and
thirdly due to challenges for both CAMS and lidar dust retrievals under conditions when dust is mixed with other aerosol types. In particular, it appears likely that CAMS underestimates dust concentrations despite the assimilation of AOD from satellites. Better observations of both dust properties such as size, composition, and optical properties as well as more widespread comparisons between CAMS (and other composition reanalyses) and lidar vertical profiles of dust on a wider scale are required.


Finally, we examined opportunities to mitigate engine dust damage by reducing dust dose. Due to the diurnal cycle in dust concentration and vertical distribution, dust concentrations peak at low altitudes in the late afternoon and are at a minimum during the night. This is particularly evident in airports with low altitude dust such as Dubai, where dust dose can be reduced by up to 41% by flying at night. Many airports saw a reduction of at least 30% from night flying. Thus changing flight times
could be a useful tool towards reducing dust dose. Variations in hold altitude can also significantly reduce total dose depending on the altitude of the dust maxima. Since dust generally decreases with altitude, raising hold altitude reduces dust dose. For example, at Delhi in JJA total dust dose can be reduced by 41%. However, knowledge of the dust vertical profile is crucial, since airports with elevated dust plumes can incur higher dose if holding altitude is raised into the dust plume. In this context, installation of airport-based ground lidars would be extremely useful in informing air traffic control services about
the presence, concentration and altitude of dust and potential manoeuvres to avoid the highest dust concentrations. Additionally, there may be costs associated with adjusting holding altitudes and opportunities to do so will vary by airport and air traffic situations. CALIOP data were not used to investigate variation in diurnal dose due to low sampling frequency. Further investigation of the diurnal variation in vertical dust profile from observations could be probed using the Cloud-Aerosol Transport System (CATS) onboard the International Space Station (ISS) (Yorks et al., 2016), though less than three
years of data are available (Lu et al., 2023).

We note that we analyse data from the CAMS *reanalysis* here, which differs somewhat from the CAMS operational forecast, which undergoes more frequent updates and improvements. Here the use of the CAMS reanalysis was most appropriate due to its long, consistent dataset for generating a climatology of dust. Nevertheless, an evaluation of operational dust models in
light of engine dust ingestion would be useful for future research, particularly since the representation of the dust life cycle in CAMS has been reviewed and improved (Remy et al., 2022). The potential use of dust forecasts operationally to adjust hold altitude for dust dose reduction could be extremely valuable.



This work provides a first quantification of aircraft engine dust dose at worldwide airports over a climatological time period.
This is an increasingly important metric, given increased aircraft operations in dusty regions such as the Middle East and technological engine developments rendering them more sensitive to dust impacts. This study also provides a framework to assess the impacts of dust on aircraft engines in other contexts, such as under climate change, where although increased likelihood of drought may exacerbate dustiness (Ukkola et al., 2020; Aryal and Evans, 2021) the latest generation of climate models show a wide disparity of present day dust emissions (Zhao et al., 2022). Finally, this work emphasizes the
importance of agreement between model and satellite datasets, not just in total column aerosol or dust optical depth, but for the vertical variation in dust, which is crucially important in determining aircraft engine dust dose.

## 7    Data availability

CAMS data are available from the Atmospheric Data Store at https://ads.atmosphere.copernicus.eu/cdsapp#!/dataset/cams-global-reanalysis-eac4?tab=overview

CALIOP                 L3                 data                 are                 available                 from
https://doi.org/10.5067/CALIOP/CALIPSO/CAL_LID_L3_Tropospheric_APro_cloudfree-Standard-V4-20

For the production of LIVAS dataset, CALIPSO data (provided by NASA) were obtained from the ICARE Data Center (http://www.icare.univ-lille1.fr/). The LIVAS dust products are available upon request from Vassilis Amiridis (vamoir@noa.gr), Emmanouil Proestakis (proestakis@noa.gr), and/or Eleni Marinou (elmarinou@noa.gr).


## 8    Author Contribution

CR, HD and RC designed the concept and experiments. CB carried out the data analysis, with guidance from CR, HD and RC. VA, EM and EP derived and provided the LIVAS data and advised on its application. SR, ZK, AB and MP advised on using the CAMS data. MV advised on application of the CALIOP data. CR prepared the manuscript, with contributions from
all co-authors.

## 9    Competing Interests

The authors declare that they have no conflict of interest.

## 10    Acknowledgements

CR was funded by NERC IRF grant number NE/M018288/1. CALIOP L3 data were obtained from the NASA Langley
Research Center Atmospheric Science Data Center. We thank the ICARE Data and Services Center for providing access to



the data used in this study and their computational center. EP was supported by AXA Research Fund for postdoctoral researchers under the project entitled "Earth Observation for Air-Quality – Dust Fine-Mode - EO4AQ-DustFM".



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
