# Peer review of "Aircraft Engine Dust Ingestion at Global Airports"

_EGUsphere, 2023_

## Author Comment (AC1)

**Review 1**

This is a neat paper, a very practical application of data and modeling efforts that the authors and others have been pursuing for years. The study reaches clear recommendations, which is especially gratifying.

We thank the reviewer for taking the time to read and review our paper. We are very pleased to hear their enthusiasm for the findings and recommendations.

Lines 183-184. This is not a complete sentence.

Thank you – now complete.

Clearly the near-surface dust concentration is especially important, and I know that CALIPSO sensitivity tends to diminish within the lowest 75 – 100 m of the surface. Is the reason that CAMS produces lower dose than CALIPSO for near-surface dust concentrations the assumption that the CALIPSO extinction at 100 m is extrapolated to the surface? The confidence with which you can assess the elevation of a near-surface concentration peak seem especially relevant based, e.g., on Figures 4c, 4e, 5c, and 5e, and it comes up again in the discussion of Figure 8. (I now see some discussion in lines 495-502. And I agree it is surprising in light of CALIOP being lower than other measurements. I'm wondering whether there is any EarliNet data that might help here.)

In terms of EARLINET sites, these cover Europe and therefore cannot be used to validate our profiles in this study. However, the suggestion of ground-based lidar validation is valuable. In fact this is the topic of ongoing research, and will be published separately in due course. We discuss the potential application of ground-based lidar data in the conclusion. As the reviewer points out we discuss these differences in the lowest portion of the vertical profile in the discussion. We have added a signposting sentence to the beginning to the results section so the reader knows the differences are discussed later on, "In this section we present the key results and findings of this work. A full discussion of the potential causes for differences between datasets is given in Section 5."

Section 2.3.3. I think you do as good a job as one can with available satellite data in constraining the dust mass concentration. But it might be worth also making some assessment of the uncertainty in mass concentration, e.g., associated with the uncertainty in the MEC. (I realize you compare the model and two lidar estimates with each other, which might represent a rough estimation of uncertainty due to s in Equation 1, but I think you are using the same MEC to obtain concentrations for all three.) (Again, I now see you say a bit more about MEC uncertainty in lines 503-513. Ok, so is the recommendation that we need better measurements of dust MEC?)

Yes, we certainly agree that we need better measurement constraints on dust MEC. Additional text has been added to the conclusions to emphasize this, "(particularly mass extinction coefficient, which is a crucial property in relating model-calculated mass loading to satellite-derived optical retrievals)."

Further, is there any available data on actual engine wear, that might be used to test or constrain the overall results?

The simple answer to the question is no, we don't have sufficient understanding of engine damage rates, relative to the dust dose, to be able to back calculate the dust dose from the amount of damage. Evidence of dust damage exists, but we don't know how much dust it takes to get that level of damage. This is why we want accurate – or as accurate as possible – data on the dust in the atmosphere and thus how much dust dose relates to the damage we observe in engines, and hence the motivation for this study.

Further, engine manufacturers *are* beginning to perform controlled laboratory engine dust tests where we know the dose and can relate this to the damage observed However, these are only in their infancy due to extreme costs involved. There are also challenges involved in reproducing in-service dust conditions, and composition of dust adds an additional complexity.

We have added the following text to the introduction to provide more information:

"Mineral dust can cause engine damage through various different mechanisms. A comprehensive description is provided by Clarkson et al. (2020), who describe how dust particles cause erosion of compressor blades, vanes and of seals between engine components, causing loss of efficiency. Dust particles can melt in hot sections of gas turbines (combustors and core turbines) and deposit on surfaces, reducing aerodynamic efficiency, damaging ceramic thermal barrier coatings, and blocking aerofoil film cooling hole features. Dust particles can also enter secondary engine air systems where they restrict cooling air flow, causing overheating or component deterioration. All these processes lead to reduced efficiency and reduced component lifetime. Resultant losses in efficiency can increase aircraft fuel burn and therefore result in increased aviation emissions of greenhouse gases, thus linking mineral dust to aviation's climate impact (e.g. Lee et al., 2021). The damage done by dust depends on the altitude and power of the engine – in part since this determines the quantity of dust ingested, but also because these factors affect dust particles' behaviour during their transit through the engine."

"Reports within the aviation industry suggest that engines operating regularly in dusty airports show evidence of accelerated component deterioration (Clarkson et al. 2020). Although this evidence exists, there is a lack of knowledge of the amount of dust required to cause such damage. Although engine manufacturers are beginning to perform controlled tests where dust concentrations are known and damage is observed, these incur extremely high costs and are currently limited."

Figure 3. I know there are some seasonal and altitude differences, but if you could reorder the entries in the legend for this figure so they are generally in the order of dust concentration magnitude, it would be easier to discern the lines associated with the lowest few, which seem to be Phoenix, Hongkong, and Sydney.

We have rearranged the order of the cities in the legend as suggested.

I understand that you are effectively using seasonal background dust levels for these calculations, and I know that for some phenomena such as dust transport in general, extreme events dominate. So, I'm wondering (a) how well the limited CALIPSO sampling and the CAMS model simulations capture sporadic larger dust events, (b) whether the airports in question shut down when dust loading is unusually elevated, and (c) how the dosage for even one elevated event for which the airport might remain open might compare with the typical seasonal averages.

These are good questions.

a) For CAMS climatology (e.g. fig 3), where we represent all model events, temporal sampling is not an issue since we use all model output. In terms of how well the model reanalysis represents larger, sporadic events, this is beyond the scope of the study. However, Errera et al. (2021) find that CAMS reproduces the annual variability fairly well compared to AERONET for dust with correlations between 0.84-0.87. They also show that over the Sahara, the largest DOD underestimations are observed during summer, and related to mesoscale convective events which the model cannot reproduce. Errera et al. (2021) and Bennouna et al. (2023) also state that the CAMS reanalysis reproduced a few recent individual large Saharan dust events reasonably well, though AODs are underestimated.

For CAMS/CALIOP profile and dose comparisons, the temporal sampling potentially becomes important. Figures 4, 5 and supplement figures indicate the number of profiles included in the long-term means shown. For example, taking the Canary Islands for DJF as an example, the CALIOP L3 data includes 146 profiles over 13 years, which is fairly low fraction sampled. Order of magnitude similarities in the sampling also apply to the LIVAS data. Therefore the satellite data are clearly a small sample of the events which occurred. We have not investigated the representativity of CALIOP overpasses as a function of larger dust events. This is beyond the current scope of the work. However, it is reassuring that the CAMS data, also sub-sampled to match the CALIOP overpasses in figure 4/5, shows similar vertical structure and magnitude compared to that using all the data (figure 3), indicating that the climatologies constructed from CALIOP L3/LIVAS sampling are reasonable.

b) Airports and air traffic control have procedures for visibility reductions in place ('Low Visibility Procedures' (LVP) and 'Reduced Aerodrome Visibility Procedures'), which vary depending on how low visibility drops. In extreme events, airports have been reported to have temporarily cancelled or postponed flights due to extreme dust events. When visibility falls beneath around 400m, airports follow these procedures, which generally results in greater spacing between aircraft and therefore reduced airport capacity. We added, "Moderately dusty conditions at airports cause reduced visibility which can require greater spacing between aircraft, and thus reduce airport capacity (ICAO 2023)," to the introduction.

c) Again, this is a very interesting question, but beyond the scope of the current work. Ongoing and future work within our research groups will investigate these questions.

Line 387. The end of the sentence is missing.

Cross-reference to section 5 now included.

Line 576. Might visibility still be an issue in the vicinity of a busy airport, even after an aircraft has left the ground?

No, it isn't a problem. Visibility is a problem on the ground because of the risk of aircraft bumping into other aircraft, ground vehicles and other obstacles. Once the aircraft is in the air the only obstacles to avoid would be other aircraft, but air traffic management procedures keep aircraft a long way apart, whatever the level of visibility. We added, "it is not a concern once aircraft have taken off since air traffic control keep large distances between aircraft, whatever the visibility."

References

Bennouna, Y. et al., 2023, Validation report for the CAMS global reanalyses of aerosol and reactive trace gases: Period 2003-2022,

---

## Author Comment (AC2)

**Review 2**

The work provides valuable information about the ingestion of atmospheric mineral dust at 10 global airports by using one reanalysis dataset and two observational datasets derived from lidar measurements. The authors compare climatological and seasonal features of dust dose and find substantial differences among the datasets. These differences are discussed in detail. The research design and methodology is considered appropriate. The results are explained in detail and the presentation of the results is adequate, but the discussion of the results could be improved. Please see detailed comments below.

I recommend the article for publication in NHESS after addressing the following comments and recommendations.

We thank the reviewer for this nice summary and positive feedback on our article. Specific comments are addressed below.

Major comment:

The differences between the datasets and therefore dust dose uncertainties are rather large at some airports and I understand that it is important to explain these differences. However, a paper with the title "Aircraft engine dust ingestion at global airports" should not only discuss differences between datasets and data retrievals in the "Discussion" section. I would rather expect some discussion about the results and their implications.

We have added a sentence at the start of the Results section to state that this section presents the main findings, while the discussion goes into more detail of the differences and reasons for these. Since the reasons for the differences between datasets are complex, we consider it clearest to approach them holistically in one section of the article. Additionally, as suggested we have added a new section discussing the implications of the results (more detail below).

My suggestion is the following:

• Move the detailed information about the model and data retrievals to the supplement and condense most important information in the results or the discussion section.

We feel that the details on the model and satellite retrievals are a core part of this study. The most important findings are summarized in the 'summary and conclusions' section. The discussion section is clearly labelled with sub-sections to signpost the reader through that section, as is the model/lidar information in the methods section.

• Compare your results to that of Bojdo et al. (2020) (see comment below) and other studies if available.

We have reduced the description of Bojdo et al. (2020) in the introduction, and added a more detailed comparison of our dose results against theirs in a new section in the Discussion, "Comparison with Existing dust dose estimations."

• Discuss the model results in terms of magnitude: Since CAMS < CALIOP < AERONET < MODIS, CAMS might significantly underestimate dust concentration at several locations. What does that mean for aircraft engines?

We have added a new section to the discussion, 'Implications of Dataset Differences' where we discuss this, particularly the CAMS < both CALIOP datasets point, since CAMS is beginning to be widely used in the aviation sector. We also added an implication statement to the abstract regarding this point.

•       Discuss the implications of a jet engine core ingesting dust (text from conclusion section, lines 594 to 610). Could you give more detailed information about the specific types of damage? Is there any critical mass of dust? Please give more information, if possible.

More information has been added to the introduction to provide more detail on the specific types of damage incurred by engines due to dust (see track changes in the manuscript and response to reviewer 1).

Regarding the concept of a critical mass, or more accurately a critical dose, all the damage mechanisms now described in the introduction have a long-term impact; they don't cause rapid failures during or shortly after the dust exposure, unlike some of the high profile volcanic ash exposures. And although volcanic ash – with its high glass content – is more likely to cause very rapid build-up of deposits in the hot section, leading to compressor stall and engine shutdown, mineral dust can do this too, at high enough concentrations. But the concentrations needed are in the 100 mg/m3 range rather than 0.1-10 mg/m3 associated with very dusty conditions.

The actual mass of deposit in a turbine section needed to get a volcanic ash type failure does vary depending on specific engine type, but is about 0.5 – 1 kg, noting that to get this in the turbine the engine core needs to ingest about 10 times this, i.e. a core dose of around 5 – 10 kg – or about 1000 flights out of somewhere like Doha. And just for good measure, the concentration of the dust going into the engine would still have to be at around 50-100 mg/m3; at lower concentrations most of the deposit will have been shed before sufficient can build up. It's a complex topic that would require a separate paper.

Minor comments:

Lines 21/22: You explicitly mention Beijing's dust dose in the abstract. Why? I suggest either removing the sentence or explaining why this information is important.

This sentence has been changed to, "Dust doses are mostly largest in northern hemisphere summer for descent, with the largest at Delhi in JJA (6.6 g) followed by Niamey in MAM (4.7 g) and Dubai in JJA (4.3 g)."

Lines 65 to 72: While the work of Bojdo et al. (2020) should definitely be mentioned in the introduction, I recommend moving the description of their work including the discussion of their findings to your discussion section (see major comment above).

See response to major point above. We have added a more substantial comparison to Bojdo et al. (2020) and moved some sections of text around.

Line 101: Did the authors also investigate dust dose at the airport in Singapore? If not, please rewrite this sentence.

Corrected

Line 124, Line 661: Please consistently write "analyse" or "analyze".

Done

Line 147: Please consistently use data as singular or plural noun (plural noun: e.g., lines 116, 153, 287, 357, 657, 680; singular noun: e.g., lines 147, 187, 541, 570).

Changed to singular.

Line 157: The "Young et al. 2018" reference is missing in the reference list.

Added

Lines 183/184: I do not understand this sentence. Please clarify.

'is generated' added to the sentence.

Lines 187 to 189: Did you do this in your study or is this part of the LIVAS processing? Please clarify and add a remark.

The re-gridding was done for our study – this sentence has been reworded to clarify.

Line 203: "two products are not similar": what does "not similar" mean? How big are these differences? This sentence seems to be the condensed version of Section 5.2 (see major comment above).

We removed this sentence as it was not necessary. This section only describes the methodological differences between LIVAS and CALIOP L3. Section 5.2 is intended to discuss the impacts of the differences on the results.

Lines 219/220: "We choose to compare the profiles using mass, since this is the metric of interest to the aviation community, though we note that extinction comparisons showed the same results.": The authors could show these results in the supplement.

Since the results show the same proportional differences and vertical structures, we do not consider it necessary to include them in the supplement.

Line 223: "Dust dose is defined as the total mass (g) of dust": When using equation 2, the unit of total mass is "kg" rather than "g".

Changed to kg

Line 241: What does "lbf" mean?

Changed to 'pounds-force'

Figure 2: Please use the same y-axis for both panels. Figure caption: add "(blue)" after "altitude" and "(black solid line)" after "wcore".

Changed

Lines 265 to 267: "All airports show the highest mean dust concentrations in JJA (driven by peak solar heating and dry convection over northern hemisphere desert regions) except Beijing and Niamey.": What about Sydney, which is located in the southern hemisphere?

We moved the sentence, "Sydney, Phoenix, Hong Kong and Bangkok all display mean dust concentrations below 10 μgm-3., and are not discussed further," to the start of this paragraph to clarify that these lines do not describe these cities.

Line 272: "Sydney, Phoenix, Hong Kong and Bangkok all display mean dust concentrations below 10 μg m-3.": The authors should add here that these airports will not be discussed later on.

See above point.

Figure 4: Is the different number of CALIOP L3 and LIVAS overpasses related to the different dust products (pure-dust vs. dust also from polluted dust products)?

Yes, this is correct. CALIOP L3 data do not include polluted dust categories (see section 2.3.1). However, LIVAS data apply the CALIOP L2 'pure dust,' 'polluted dust' and 'dusty marine' categories, extracting the 'pure dust' component from them (see section 2.3.2). Therefore the number of profiles contributing to each dataset is different, with LIVAS including more profiles. Some of the consequences of this are discussed in the Discussion.

Lines 359 to 383: Compared to the description and discussion of the other figures, this text gives to much detailed information and some aspects are not clear. For example: lines 359/360: "…CALIOP L3 substantially larger than both, are most evident at airports with a low altitude dust plume, particularly Niamey in DJF/MAM and Dubai year-round." In Niamey, the DJF and MAM medians of both LIDAR datasets are in very good agreement. In Dubai, the DJF and SON medians of both LIDAR datasets are also in very good agreement. Please clarify and shorten the text.

The text has been adjusted slightly to clarify, "The largest differences, with LIVAS and CALIOP L3 values larger than CAMS, and CALIOP L3 frequently larger than LIVAS, are most evident at airports with a low altitude dust plume, particularly Niamey in DJF/MAM and Dubai year-round." And later on, "Differences between CALIOP L3 and LIVAS in these cases are most marked when the hold altitude mass concentrations differ." We believe that this figure presents the core results of our study, and that therefore the description length is justified.

Lines 389/390: "For departure (see supplement), overall the similarities and differences between the datasets are the same as for arrival, with lower doses for departure by 10 to 23%.": Do these numbers refer to the median or to the lower/upper quartile?

This refers to the median (clarified in text).

Lines 390/391: "However, in a few cases differences between datasets compared to arrival are sensitive to the overall vertical profile shape and magnitude, particularly if ground concentrations are very large." Isn't this always the case because of the location of the hold altitude?

No – here we aim to explain that generally descent dose is larger due to the hold pattern contribution to dose. However, in a few cases this is the opposite: ascent dose is larger than descent. We reworded this section to read:

"For departure (see supplement), overall the similarities and differences between the datasets are the same as for arrival, with lower median doses for departure by 10 to 23%, due to the lack of hold phase dose contribution during ascent. However, in a few cases doses are higher for departure than arrival. This is because differences between datasets are sensitive to the overall vertical profile shape and magnitude, particularly if ground concentrations are very large."

Lines 401 to 403: "This is partly a feature of the larger magnitudes seen in the lidar data compared to CAMS, but also due to the lower sampling rate for CALIPSO compared to the regular 3 hourly model output from CAMS.": Did you use all CAMS data or only CAMS data coincident with CALIOP measurements? Please clarify.

We only use CAMS data which was coincident with CALIOP L3 sampling times in Section 3.2.2 onwards (i.e. comparisons between CAMS/lidar). This has been added to section 2.2. ("When compared against spaceborne lidar data, CAMS data are restricted to overpass times for comparison purposes.") The sentence referred to by the reviewer has been deleted as it was potentially misleading, now reading, "This is partly a feature of the larger magnitudes seen in the lidar data compared to CAMS, but and perhaps an indication that CAMS does not represent infrequent, larger dust events particularly well."

Lines 437/438: "For the peak dust seasons, a reduction in dose of 41% at Dubai in JJA, 34% at Delhi in JJA and 39% at Niamey in DJF could be achieved.": I try to understand how you calculated these numbers by looking at Delhi: According to Figs. 6 and 7, maximum dust dose in JJA is 6.6 g and 4.4 g for arrival and departure, respectively. This sums up to 11 g. However, according to Table 1, the reduction of 34 % refers to 6.44 g, which indicates that total dust was approximately 19 g. Where does this difference come from? Please clarify.

The values in Table 1 refer to reductions (g) and percentage differences between dose at the *maximum* and *minimum* during the diurnal cycle, rather than the diurnal mean as shown in figures 6 and 7. Values in Table 1 are calculated from seasonal airport mean doses, separated into 3 hourly intervals, incorporating both an ascent and a descent. E.g. for Delhi in JJA, once we have doses for every 3 hours of the day, as a climatological mean, we subtract the time of day with the lowest dose from the time of day with the highest dose. This gives the reduction in dose and its percentage equivalent (here ~6.4 g reduction from a maximum of ~19g). The manuscript states, "we show the reduction in dust dose possible for each the six dustiest airports between the maximum and minimum throughout the diurnal cycle, for an arrival immediately followed by a departure, equivalent to aircraft delaying arrival and departure from late afternoon to night time."

We adjusted the caption of table 1 slightly to read, "Seasonal mean reduction in dust dose between maximum and minimum throughout the diurnal cycle for an arrival directly followed by a departure for the CAMS dataset, given in g and as a percentage."

Table 1: Please specify only one decimal place for dose reduction.

Done

Lines 484/485: "without mixing from other aerosol types (e.g., Marrakesh and the Canary Islands)": What about sea salt aerosols in Canary Islands? Please clarify.

During the peak dust season at the Canary Islands (boreal summer) the dust plume is elevated (i.e. not at the same altitude as marine aerosol) and most of the aerosol mass and optical contribution comes from dust, rather than other aerosol types.

Section 6: Conclusion: I'd suggest renaming this section to "Summary and conclusions".

Done

Lines 633/634: "resulting in a mean underestimate by CAMS of 2.4 over both datasets": I do not understand the relevance of the mean bias over both lidar datasets. I suggest removing this part of the sentence.

Agreed, this has been removed and the figures in the abstract adjusted to reflect this change.

Lines 657 to 660: I suggest moving these two sentences to line 650 and mentioning that the results discussed before were based on the CAMS reanalysis.

We feel that these sentences fit in here as they correspond to limitations/further work relating to the diurnal dust dose findings. We have added a few extra words to signpost the reader better through this paragraph in the manuscript.

Supplement, Figure S5: Canary Island and Delhi: LIVAS data are missing.

These have been included.

Editorial:

Line 21: add "northern hemisphere" before summer.

Done

Line 25: Add "instrument" after (CALIOP).

Should not be required since the sentence indicates it's a lidar.

Line 29: "up to 44% and 41% respectively" rather than "up to 44% or 41% respectively", I think.

Changed

Section 1: remove "1".

Changed

Line 49: "O'Connell" rather than "O'connell".

Changed

Line 64: "(Inness et al. 2019)" rather than "(Inness et al. (2019))".

We believe the text is correct, due to the use of 'e.g.' in the sentence. We leave this decision to ACP copy-editing.

Line 76: "CAMS forecasts and reanalyses" rather than "CAMS forecasts and reanalysis".

The CAMS reanalysis is singular.

Line 81: Add a reference after "dust properties", e.g., Mona et al. 2012, doi:10.1155/2012/356265.

We added Winker et al. (2007).

Line 83: Add "instrument" after "(CALIPO)".

The sentence indicates it is a lidar, so we do not consider this addition necessary.

Line 84: "(e.g. Liu et al. 2008a; Yang et al. 2013; Song et al. 2021)" rather than "(e.g. Liu et al. (2008a); Yang et al. (2013); Song et al. (2021))".

We believe the text is correct, due to the use of 'e.g.' in the sentence. We leave this decision to ACP copy-editing.

Line 86: "12-year period" rather than "12 year period".

Line 93: "Section 5 concludes": This is not true. Section 5 is the "Discussion" section, Section 6 concludes.

Changed

Line 99: Please exchange "Bejing" and "Bangkok" to follow the numbering.

Done

Line 100: "'dust belt'," rather than "'dust belt,'".

Commas should be placed inside speech marks.

Line 103/104: add "coarse" before "model resolution" and refer to Section 2.2.

Done

Line 113: Introduce "ECMWF".

Done

Line 121: "lower/upper diameters" rather than "upper/lower diameters".

Done

Line 141/142: I suggest moving the sentence "The CALIPSO orbit track…." to line 134 (after the Winkler et al. (2010) reference.

Done

Line 142: Add a break before "Here we analyze…".

Done

Line 158: "reported by Floutsi et al. (2022)" rather than "reported by (Floutsi et al., 2022)".

Changed

Line 160: add "latitude-longitude" before "grid".

Done

Line 161: Add a break before "We use the extinction coefficient".

Done

Line 161: "extinction coefficient profiles at 532 nm" rather than "extinction coefficient at 532 nm profiles".

Done

Lines 165, 207: "CALIOP" rather than "CALIPSO".

Done

Line 166: "an altitude of 90 m" rather than "an altitude 90 m".

We feel the wording is acceptable here.

Line 172: (LIVAS, Amiridis et al., 2015) rather than "(LIVAS, Amiridis et al. (2015))".

This is an in-sentence citation due to the word LIVAS and cited as author (year).

Line 174: "EARLINET, Pappalardo et al., 2014; last" rather than "EARLINET, Pappalardo et al. (2014); last".

Done

Lines 180 and 197: "'pure dust', 'dusty marine'," rather than "'pure dust,' 'dusty marine,'".

Commas should be placed inside inverted commas.

Line 191: add "as input" after "profiles".

Done

Line 202: "sr" rather than "Sr".

Done

Lines 202/203: "(Amiridis et al., 2013; Marinou et al., 2017; Proestakis et al., 2018; Floutsi et al., 2022, see supplement)" rather than "((Amiridis et al., 2013; Marinou et al., 2017; Proestakis et al., 2018; Floutsi et al., 2022), see supplement)".

We leave this decision for copy-editing.

Line 224: "from Clarkson (2020)" rather than "from (Clarkson, 2020)".

Done

Line 226: add "(dimensionless)" after "regime".

Done

Line 235: "thermodynamics" rather than "theromodynamics".

Done

Line 252: "discuss" rather than "test".

We leave this as 'test' since we do indeed test the sensitivity.

Line 301: "are" rather than "were", I think.

Done

Line 304: "show" rather than "showed", I think.

Done

Line 313: "greater than 2 g in winter": it is even greater than 3 g in winter.

That is correct, but not the point we are making.

Line 316: "contributes to at least 50%" rather than "contributes at least 50%", I think.

Done

Line 350: remove "to those calculated from".

Sentence adjusted.

Figure 8: Please use the same y-axis for all panels; please capitalize the first letter of each airport name. Does it make sense to use an exponential y-axis?

These changes have been made. We changed the y-axis to a semi-log axis to make better use of the figure space.

Line 387: The Section number is missing.

Done

Line 433: "for the six dustiest airports" rather than "for each airport".

Done

Line 438: "an" rather than "a an".

Done

Line 439: Maximum reduction is 19 % in Canary Islands.

Changed

Line 482: remove "the reason" after "explain".

Done

Line 489: "(e.g., Zhao et al., 2022)" rather than "(e.g., Zhao et al. (2022))".

We believe the text is correct, due to the use of 'e.g.' in the sentence. We leave this decision to ACP copy-editing.

Lines 495/496: "(e.g., Schuster et al., 2012; Sayer et al., 2018; Song et al., 2021)" rather than "(e.g. Sayer et al. (2018); Schuster et al. (2012); Song et al. (2021)".

We believe the text is correct, due to the use of 'e.g.' in the sentence. We leave this decision to ACP copy-editing.

Line 500: "DOD" rather than "DAOD".

Changed

Line 509: "O'Sullivan" rather than "O'sullivan".

Changed

Line 526: "sr" rather than "Sr".

Done

Line 578: Add a break before "This study…".

Done

Line 599: "representative of the region around" rather than "representative of region around".

Done

Line 613: "spent there, which" rather than "spent in the hold pattern, which".

We feel this wording makes the meaning of the sentence clear.

Line 625: "the vertical structure at Beijing" rather than "the vertical structure shape at Beijing".

Done

Line 642; remove "both" before "dust properties".

Done

Line 652: add "by increasing hold altitude from 1 km to 3 km" after "can be reduced by 41%".

This figure does not refer to hold altitude change, therefore we leave the sentence unchanged.